# Non-conventional octameric structure of C-phycocyanin

Takuo Minato [1,2,3,9], Takamasa Teramoto [4,9], Naruhiko Adachi [5], Nguyen Khac Hung[1,2], Kaho Yamada[1,2], Masato Kawasaki[5,6], Masato Akutsu[5], Toshio Moriya [5], Toshiya Senda [5,6], Seiji Ogo[1,2,7], Yoshimitsu Kakuta [4,8 ✉] & Ki-Seok Yoon [1,2,7 ✉]

C-phycocyanin (CPC), a blue pigment protein, is an indispensable component of giant phycobilisomes, which are light-harvesting antenna complexes in cyanobacteria that transfer energy efficiently to photosystems I and II. X-ray crystallographic and electron microscopy (EM) analyses have revealed the structure of CPC to be a closed toroidal hexamer by assembling two trimers. In this study, the structural characterization of non-conventional octameric CPC is reported for the first time. Analyses of the crystal and cryogenic EM structures of the native CPC from filamentous thermophilic cyanobacterium *Thermoleptolyngbya* sp. O–77 unexpectedly illustrated the coexistence of conventional hexamer and novel octamer. In addition, an unusual dimeric state, observed via analytical ultracentrifugation, was postulated to be a key intermediate structure in the assemble of the previously unobserved octamer. These observations provide new insights into the assembly processes of CPCs and the mechanism of energy transfer in the light-harvesting complexes.

[1] Department of Chemistry and Biochemistry, Graduate School of Engineering, Kyushu University, 744 Moto-oka, Nishi-ku, Fukuoka 819-0395, Japan. [2] International Institute for Carbon-Neutral Energy Research (WPI-I2CNER), Kyushu University, 744 Moto-oka, Nishi-ku, Fukuoka 819-0395, Japan. [3] Department of Applied Chemistry, Graduate School of Advanced Science and Engineering, Hiroshima University, 1-4-1 Kagamiyama, Higashi-Hiroshima, Hiroshima 739-8527, Japan. [4] Department of Bioscience and Biotechnology, Faculty of Agriculture, Kyushu University, 744 Moto-oka, Nishi-ku, Fukuoka 819-0395, Japan. [5] Structural Biology Research Center, Institute of Materials Structure Science, High Energy Accelerator Research Organization (KEK), 1-1 Oho, Tsukuba, Ibaraki 305-0801, Japan. [6] Department of Materials Structure Science, School of High Energy Accelerator Science, The Graduate University of Advanced Studies (Soken-dai), 1-1 Oho, Tsukuba, Ibaraki 305-0801, Japan. [7] Center for Small Molecule Energy, Kyushu University, 744 Moto-oka, Nishi-ku, Fukuoka 819-0395, Japan. [8] Laboratory of Structural Biology, Graduate School of System Life Sciences, Kyushu University, 744 Moto-oka, Nishi-ku, Fukuoka 819-0395, Japan. [9] These authors contributed equally: Takuo Minato, Takamasa Teramoto. ✉email: kakuta@agr.kyushu-u.ac.jp; yoon@i2cner.kyushu-u.ac.jp

light-harvesting in photosynthesis is a crucial initial step in the conversion of light energy into chemical energy;[1,2] therefore, structural investigation of light-harvesting complexes is important for gaining an understanding of energy transfer mechanisms and developing effective systems of artificial photosynthesis[3–5]. Phycobilisomes (PBSs) are light-harvesting, antenna complexes that are located on the thylakoid membranes of cyanobacteria, red algae, and glaucophytes; they play a key role in the transfer of energy to photosystems I and II[6–9]. The most common hemidiscoidal-type PBSs consist of core and peripheral rod moieties; the core consists of cylindrical substructures of allophycocyanins (APCs), whereas the rod consists of stacked phycocyanins (PCs) and phycoerythrins (PEs)[10]. These phycobiliproteins (PBPs) possess different types of open-chain tetrapyrrole chromophores covalently linked by thioether bonds, resulting in the different absorption properties; the maximum absorptions in APC, PC, and PE are observed at approximately 650–655, 615–620, and 540–570 nm, respectively, which enables PBSs to increase the range of photon absorption from sunlight[7,8]. Recently, cryogenic electron microscopy (cryo-EM) analyses have revealed the complete structures of the PBSs from *Griffithsia pacifica* and *Porphyridium purpureum* in high resolution; these analyses illustrate that the interaction between the aromatic amino acids of the linker proteins and the chromophores is likely to be a key factor in the efficient transfer of energy[11,12]. Consequently, the well-organized geometrical arrangement of PBPs and linker proteins in PBSs show substantially high overall quantum efficiency (>90%)[13].

Structurally, PBPs consist of α and β subunits that form the (αβ) monomer; three monomers are then assembled into an (αβ)$_3$ trimer, which is the basic unit of the PBP[8]. C-phycocyanin (CPC), a blue pigment PC mainly observed in cyanobacteria, is a key component in the transfer of energy from the rod to the core[7]. X-ray crystallographic analyses and EM analyses of CPCs have revealed that the quaternary structure of CPC is a closed toroidal (αβ)$_6$ hexamer with $D_3$ symmetry that is thought to be formed by assembling two trimers face-to-face[11,12,14–27]. In solutions, however, the hexameric (αβ)$_6$ state of CPC is only stable in a concentrated phosphate buffer solution at neutral or low pH; it disassembles into trimeric (αβ)$_3$ or even monomeric (αβ) states at low phosphate buffer or low CPC concentrations[28,29]. Therefore, CPC-containing solutions isolated by conventional methods, i.e., column chromatography, usually contain a mixture of monomers, trimers, and hexamers depending on the pH and concentrations of phosphate and CPC[28–30]. Nevertheless, reversible disassembly/reassembly reactions between monomers, trimers, and hexamers have been confirmed in vitro, mainly by analytical ultracentrifugation;[31] thus, it has been suggested that the monomer–trimer–hexamer equilibrium is likely involved in the assembly process of CPCs in vivo[8,23,32]. According to studies on the biosynthetic pathways of CPC, the early events in PBS assembly are also based on the assumption that CPCs have a propensity to form hexameric (αβ)$_6$ states via trimeric (αβ)$_3$ and monomeric (αβ) states[15,28,33].

We recently isolated a filamentous thermophilic cyanobacterium *Thermoleptolyngbya* sp. O-77 (O-77) from a hot spring in Japan[34–36]. During the course of a study on the photosynthesis of O-77, we isolated a thermostable photosystem II complex from O-77[34]. Thus, we now focus on the isolation of PBPs from O-77 to investigate their light-harvesting properties. In the present study, the native CPC from O-77, namely *Tl*CPC, was isolated and its oligomeric states were determined using X-ray crystallographic and cryo-EM analyses. These structural analyses unexpectedly revealed the coexistence of conventional hexameric (αβ)$_6$ (*Tl*CPC-6) and novel octameric (αβ)$_8$ (*Tl*CPC-8) states, even though *Tl*CPC-6 and *Tl*CPC-8 possessed the same amino

acid sequences; notably, PBPs with $C_4$ or $D_4$ symmetries have not previously been reported. In addition, analytical ultracentrifugation of the *Tl*CPC solution showed that the main dissolved species was an unusual dimeric (αβ)$_2$ state. Together with the dissociation free energies of oligomeric (αβ)$_n$ states ($n = 2, 3, 4, 6$, and 8), our experimental results indicate that the previously unobserved octamer was presumably assembled from four dimers. To the best of our knowledge, *Tl*CPC-8 is not only the first example of an octameric CPC but also the first reported assemblage of identical subunits into different cyclic oligomers with 3-fold and 4-fold rotation axes.

## Results and discussion

**Isolation and characterization of *Tl*CPC.** Based on genome sequence analysis, O-77 possesses CPC-, APC-, and linker protein-encoding genes but entirely lacks PE-encoding genes, which is similar to several cyanidophytina and cyanobacteria[37]. The genome encodes single genes for *CpcA* (BAU42407) and *CpcB* (BAU42408), encoding the α and β subunits for CPC. Since the *ApcE* gene in O-77 (BAU42084), which encodes the α subunit of APC with a linker in the core of PBS, contains four repeat domains (pfam00427 domains in the Pfam database)[6,38], the PBS complex in O-77 is expected to possess pentacylindrical APC cores[39,40]. It should be noted that a whole structure of this PBS type has, to date, rarely been investigated at high resolution[24,41].

Initially, *Tl*CPC was isolated using a conventional column chromatography technique (see Methods section for details). In SDS-PAGE of *Tl*CPC, approximately 17.5-kDa and 22.5-kDa bands, assignable to the α and β subunits of *Tl*CPC, respectively, were observed (Supplementary Fig. 1). The observed 17.5-kDa band agreed well with the calculated mass of the α subunit (17.4 kDa), whereas the 22.5 kDa band was larger than expected (β subunit: 17.9 kDa). The UV–vis spectrum of *Tl*CPC showed absorption bands at 277, 353, and 615 nm, while the fluorescence spectrum showed an emission band at 644 nm; these are characteristic bands of CPC (Supplementary Fig. 2). The circular dichroism spectrum of *Tl*CPC showed negative Cotton effects with maxima at 220 and 209 nm and a positive Cotton effect with a maximum at 193 nm, which are characteristic bands of α-helix structures (Supplementary Fig. 3). These results were compatible with those previously reported for CPCs and suggested that *Tl*CPC had been successfully isolated[7,14,25].

**X-ray crystallographic analysis of *Tl*CPC-6.** To elucidate the oligomeric state of *Tl*CPC, the crystallization of *Tl*CPC was examined under various crystallization conditions, which resulted in the formation of many blue single crystals (Supplementary Table 1), where crystal systems were determined in the early data processing stages. The initial screening of X-ray crystallographic analyses revealed that single crystals belonged to the space groups of $P2$ (No. 2), $P2_1$ (No. 4), and $P2_12_12$ (No. 18), which represented the conventional hexameric (αβ)$_6$ states. *Tl*CPC crystallized in the primitive orthorhombic $P2_12_12$ was further refined and then the hexameric structure (*Tl*CPC-6) with $D_3$ symmetry was determined at a resolution of 1.65 Å (Fig. 1a, c; Table 1), where the asymmetric unit contained a hexamer. In a comparison with the crystal model of *Spirulina platensis* CPC (*Sp*CPC; PDB ID: 1GH0), the crystals of which contained two hexamers in the asymmetric unit[17], the root mean square deviation (rmsd) value was 0.68 Å over 1,993 Cα atoms, supporting the formation of a well-established hexameric (αβ)$_6$ state. The α and β subunits of *Tl*CPC-6 shared 68–86% and 65–87% sequence identities, respectively, with the previously reported crystal models of CPCs[14–27]. The X-ray crystallographic analysis of *Tl*CPC-6 showed that phycocyanobilins (PCBs), i.e., the chromophores in

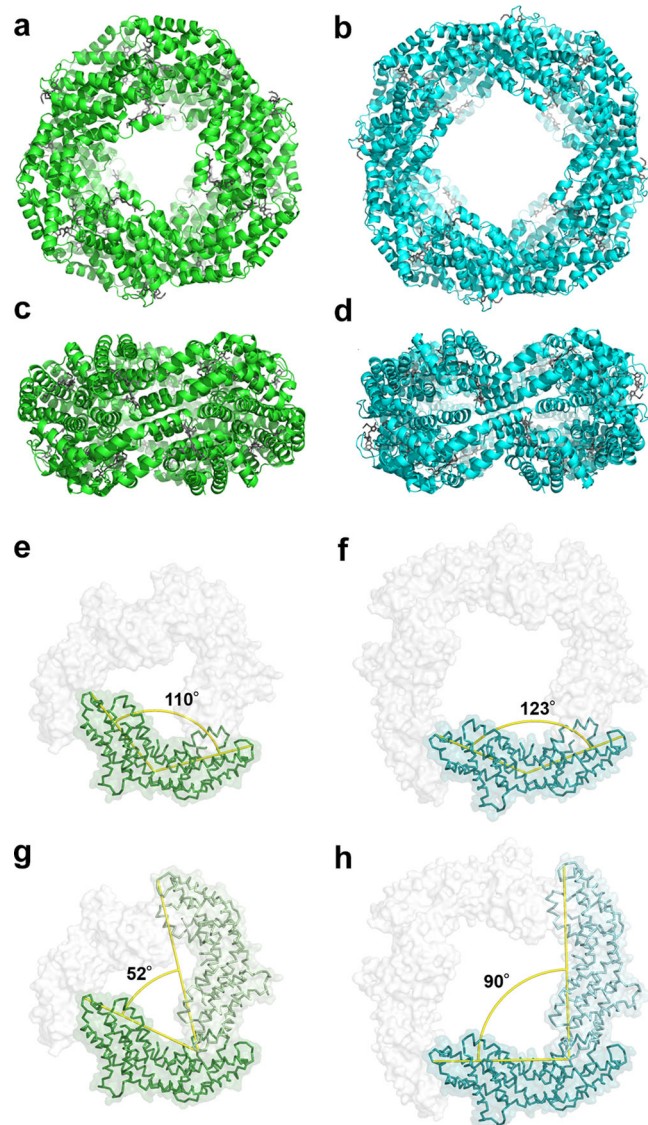

**Fig. 1 Crystal models of *Tl*CPC-6 and *Tl*CPC-8.** Top and side views of *Tl*CPC-6 (**a**, **c**) and *Tl*CPC-8 (**b**, **d**). Dihedral angles of monomers in *Tl*CPC-6 (**e**) and *Tl*CPC-8 (**f**). Angles between adjacent monomers in *Tl*CPC-6 (**g**) and *Tl*CPC-8 (**h**). The main and side chains of *Tl*CPC-6 and *Tl*CPC-8 are represented by green and cyan ribbon models, respectively. PCBs are represented by grey lines. Yellow lines are guides for measuring angles.

**Table 1 Crystallographic data collection and refinement statistics.**

| (PDB ID) | *Tl*CPC-6 (7EFW) | *Tl*CPC-8 (7EFV) |
|---|---|---|
| ***Data collection*** | | |
| Space group | $P22_12_1$ (No. 18) | $I432$ (No. 211) |
| ***Cell dimensions*** | | |
| $a, b, c$ (Å) | 60.1, 187.4, 210.1 | 230.0, 230.0, 230.0 |
| $\alpha, \beta, \gamma$ (°) | 90.0, 90.0, 90.0 | 90.0, 90.0, 90.0 |
| Resolution (Å) | 50.0-1.65 (1.76-1.65)* | 50.0-2.77 (2.93-2.77)* |
| $R_{means}$ | 21.2 (106.2) | 37.8 (420.0) |
| $CC_{1/2}$ | 99.3 (49.5) | 99.8 (55.7) |
| $I/\sigma I$ | 9.5 (1.53) | 11.39 (0.91) |
| Completeness (%) | 92.6 (65.1) | 100.0 (100.0) |
| Redundancy | 5.8 (2.9) | 61.6 (61.7) |
| ***Refinement*** | | |
| Resolution (Å) | 49.18-1.65 | 49.03-2.77 |
| No. reflections | 264075 | 26620 |
| $R_{work}/R_{free}$ (%) | 17.0/19.0 | 19.9/23.8 |
| ***No. atoms*** | | |
| Protein | 15354 | 5021 |
| Ligand | 774 | 258 |
| Water | 2676 | — |
| ***B-factors*** | | |
| Protein | 16.4 | 68.8 |
| Ligand | 16.8 | 68.9 |
| Water | 29.3 | — |
| ***R.m.s. deviations*** | | |
| Bond lengths (Å) | 0.005 | 0.003 |
| Bond angles (°) | 0.985 | 0.829 |
| ***Ramachandran plot*** | | |
| Favored (%) | 98.3 | 98.0 |
| Allowed (%) | 1.7 | 2.0 |
| Disallowed (%) | 0 | 0 |

*Values in parentheses are for a highest-resolution shell.

CPCs, were covalently bonded at the conserved Cys residues of αCys84 (α84PCB), βCys82 (β82PCB), and βCys153 (β153PCB) (Supplementary Fig. 4). The conformations of PCBs were essentially identical to those in CPCs from *Synechococcus elongatus* (*Se*CPC; PDB ID: 1JBO)[19], which were previously determined at high resolution (1.45 Å) with a low $R_{free}$ value (Supplementary Fig. 5). The identical conformations of aromatic rings in PCBs also supported the typical absorption and emission spectra observed in *Tl*CPC (Supplementary Fig. 2). Methylation of βAsn72 is a highly conserved post-translational modification in all reported CPCs with few exceptions[20,26] and is thought to play a crucial role in highly efficient energy transfer[42]. In *Tl*CPC-6, the methyl group at the βAsn72 residue could be clearly modeled into an electron density map (Supplementary Fig. 6).

Determining the crystal packing of CPCs is important for estimating the intra- and inter-CPC energy transfer pathways because orientationally aligned flat-shaped CPCs that are assembled face-to-face in crystals can be regarded as a motif of layered CPCs in the rod moieties of PBSs, even in the absence of linker proteins[15,21,22]. Approximately 90% of the previously reported single crystals from CPCs belong to the space groups of $P2_1$ (No. 4), $R32$ (No. 155), and $P6_3$ (No. 173), and most showed orientationally aligned hexameric CPCs in crystals. Only two previous reports indicated that CPCs were crystallized in the orthorhombic space group, with these crystals containing two types of orientation[27,30]. Although *Tl*CPC-6 was also crystallized in the primitive orthorhombic space group ($P2_12_12$), hexameric CPCs were orientationally aligned in the crystal. In *Tl*CPC-6, the buried surface area between hexamers along the $C_3$ axis, as calculated by the PISA program[43], was 609 Å² per hexamer, which indicated weak interactions similar to layered CPCs in the rods of PBS (Fig. 2a). Rod-like layered structures of orientationally aligned CPCs, of which $C_3$ axes are shared, have also been observed in CPCs from *Microchaete diplosiphon* (*Md*CPC; PDB ID: 1CPC)[14], *Thermosynechococcus vulcanus* (*Tv*CPC; PDB ID: 1KTP)[18], and *Thermosynechococcus elongatus* BP-1 (*Te*CPC; PDB ID: 3L0F). In *Tl*CPC-6, the distances of intraprotein PCB pairs were as follows: α84PCB–β82PCB (20.0 Å), β153PCB–β153PCB (25.8 Å), and α84PCB–α84PCB (26.9 Å) (Supplementary Fig. 7). Additionally, the distances of interprotein PCB pairs were as follows: β82PCB–β82PCB (26.2 Å) and α84PCB–α84PCB (34.4 Å). These values were close to those of the reported CPCs, indicating that the estimated main energy pathways between hexamers were via α84PCB and β82PCB. It is noteworthy that the structure of a double-layered $[(\alpha\beta)_6]_2$ unit in *Tl*CPC-6 crystal closely resembles that of a double-layered PE–PC unit in PBS from *P. purpureum* (PDB ID: 6KGX) (Supplementary Fig. 8),

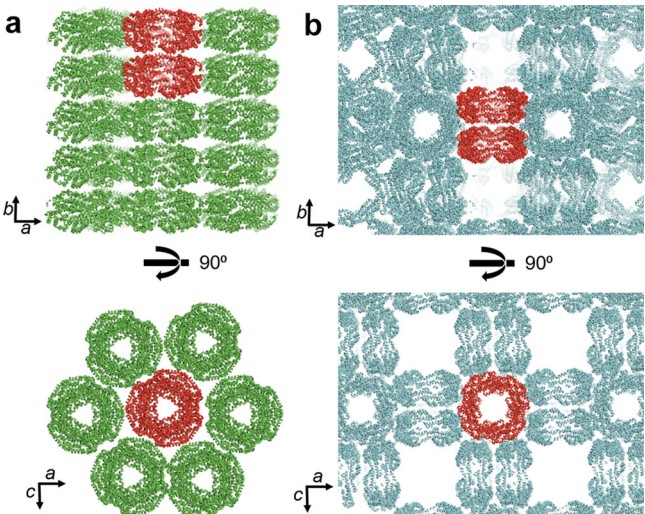

**Fig. 2 Crystal packings of *Tl*CPC-6 and *Tl*CPC-8.** *Tl*CPC-6 (**a**) and *Tl*CPC-8 (**b**) in crystals are shown as ribbon representations. Two representatives rod-like $[(\alpha\beta)_6]_2$ and $[(\alpha\beta)_8]_2$ structures are shown in red.

supporting that layered CPCs in crystals are structurally correlated with rod moieties of PBSs in vivo.

Taken together, the aforementioned results show that *Tl*CPC crystallized in the space group $P2_12_12$ was a conventional hexameric $(\alpha\beta)_6$ structure with a layered crystal packing; thus, the distinctive features of *Tl*CPC-6 were hardly observed in comparison with previously reported CPCs.

**X-ray crystallographic analysis of *Tl*CPC-8.** During X-ray crystallographic analyses, we unexpectedly found crystals that belonged to the extraordinary space group of *I* centered cubic *I*432 (No. 211) (Supplementary Table. 1), in which CPCs crystallized in a cubic space group have not previously been reported. X-ray crystallographic analysis revealed that the structure of *Tl*CPC in the space group of *I*432 at a resolution of 2.77 Å was not a conventional hexamer but rather a novel octamer (*Tl*CPC-8) (Fig. 1b, d; Table 1). This crystal model contained a dimer $(\alpha\beta)_2$ in the asymmetric unit and consisted of eight $(\alpha\beta)$ monomers to give an extended closed toroidal structure with $D_4$ symmetry. To date, all previously reported PBPs, including APCs, PCs, and PEs, have consisted of basic trimeric $(\alpha\beta)_3$ units with $C_3$ symmetry; in contrast, *Tl*CPC-8 possessed a 4-fold rotation axis that could not be directly assembled from trimers or hexamers with 3-fold rotation axes. Notably, *Tl*CPC-8 is the first example of a PBP with a 4-fold rotation axis as well as an octameric CPC.

The inner diameter of ring-shaped *Tl*CPC-8 (45 Å) increased in comparison with that of *Tl*CPC-6 (25 Å) due to the extended toroidal structure of *Tl*CPC-8 (Fig. 1a, b). The structural differences in the $(\alpha\beta)$ monomer unit between *Tl*CPC-6 and *Tl*CPC-8 (rmsd value of 2.06 Å over 321 Cα atoms) were larger than those between *Tl*CPC-6 and *Tv*CPC (rmsd value of 0.33 Å over 329 Cα atoms) due to the slight structural conformation changes. In the $(\alpha\beta)$ monomer, the helices A, B, E, and F in the α subunit ($A_\alpha$, $B_\alpha$, $E_\alpha$, and $F_\alpha$), the helices A, B, C, and E in the β subunit ($A_\beta$, $B_\beta$, $C_\beta$, and $E_\beta$), and the loops between these helices ($A/B_\alpha$, $E/F_\alpha$, $A/B_\beta$, and $B/C_\beta$) interacted with each other to form a bent structure (Supplementary Fig. 9). The dihedral angle of the $(\alpha\beta)$ monomer (αGln70–αArg30–βGly70) in *Tl*CPC-6 was approximately 110°, whereas the equivalent angle in *Tl*CPC-8 was approximately 123° (Fig. 1e, f). In addition, the angle between two adjacent monomers (αGln70–βPro69–βPro69) in *Tl*CPC-6 was approximately 52°, whereas the equivalent angle in *Tl*CPC

was approximately 90° (Fig. 1g, h). These results indicate that the structural flexibilities of the subunits and their interfaces presumably affect the assembly process of hexameric $(\alpha\beta)_6$ and octameric $(\alpha\beta)_8$ states even though both crystal models possess the same amino acid sequence.

In the crystal of *Tl*CPC-8, two octameric $(\alpha\beta)_8$ units were stacked along the $C_4$ axis to form a double-layered orientationally aligned structure of $[(\alpha\beta)_8]_2$, which were further assembled at right angles to each other. Consequently, the overall crystal model of *Tl*CPC-8 possessed a unique framework with exceedingly large void spaces of approximately $100 \times 100 \times 100$ Å$^3$ (Fig. 2b; Supplementary Fig. 10). From an engineering perspective, the topology of this structure resembles zeolite A as well as several types of metal-organic frameworks and polyoxometalates[44,45]; thus, the single crystals of *Tl*CPC-8 could potentially be utilized as catalysts and adsorbents to take advantage of the expected large specific surface area.

Considering $[(\alpha\beta)_8]_2$ units in the crystal, the layered structures can be regarded as octameric versions of rods (Fig. 2b). The overall conformations of PCBs in *Tl*CPC-8 were essentially identical to those in *Tl*CPC-6 (Supplementary Fig. 5). In *Tl*CPC-8, the distances of intraprotein PCB pairs were as follows: α84PCB–β82PCB (20.7 Å), β153PCB–β153PCB (21.7 Å), and α84PCB–α84PCB (31.6 Å) (Supplementary Fig. 11). These values were similar to those in *Tl*CPC-6, whereas the distance of interprotein PCB pairs of β82PCB–β82PCB (31.7 Å) was increased relative to the equivalent distance in *Tl*CPC-6 (26.2 Å) because a contact area was not observed between octamers by the PISA program, presumably due to the demand for octamers to fit the unique crystal packing of *Tl*CPC-8. Nevertheless, the distances of the interprotein PCB pairs were similar to the shortest distances between chromophores in PE–PE (24.4 Å) and PE–PC (23.8 Å) of *Pp*PBS, indicating the potential intra- and inter-protein energy transfer in *Tl*CPC-8.

As the crystal shapes of *Tl*CPC-6 and *Tl*CPC-8 were different from each other (Supplementary Fig. 12), CPC solutions derived from hexameric $(\alpha\beta)_6$ and octameric $(\alpha\beta)_8$ states could be prepared by picking up crystals and dissolving them separately. Matrix-assisted laser desorption/ionization time-of-flight (MALDI-TOF) mass spectrometric analysis of these two solutions showed essentially the same two signals assignable to the α and β subunits of *Tl*CPC (Supplementary Fig. 13). In addition, CD spectra of these solutions also showed almost the same ellipticity in a range of 190–250 nm (Supplementary Fig. 14a), and the thermal denaturation midpoints were 71 °C for both solutions (Supplementary Fig. 14b). It should be noted that oligomeric structures were disassembled in these solutions because of the low concentration of CPC[28–32]. These results strongly supported that the subunits composition and physical properties of *Tl*CPC-6 and *Tl*CPC-8 were the same and post-translational modifications did not occur except for the methylation and PCB chromophorylation (Supplementary Figs. 5 and 6). Crystals seemed to be grown one type of morphology in the same drop, and the residual crystallization solution were much clearer than the initial states (Supplementary Fig. 12), indicating the equilibrium between hexameric $(\alpha\beta)_6$ and octameric $(\alpha\beta)_8$ states during the crystallization process. Collectively, the aforementioned results confirm that the native *Tl*CPC was crystallized in two different morphologies, *Tl*CPC-6 and *Tl*CPC-8, from the same monomeric $(\alpha\beta)$ units. Moreover, the coexistence of both structures in concentrated solutions is strongly suggested.

**Cryo-EM structure of *Tl*CPC.** To avoid the effect of crystallization, the presence of hexameric $(\alpha\beta)_6$ and octameric $(\alpha\beta)_8$ states was further investigated via cryo-EM analysis. The

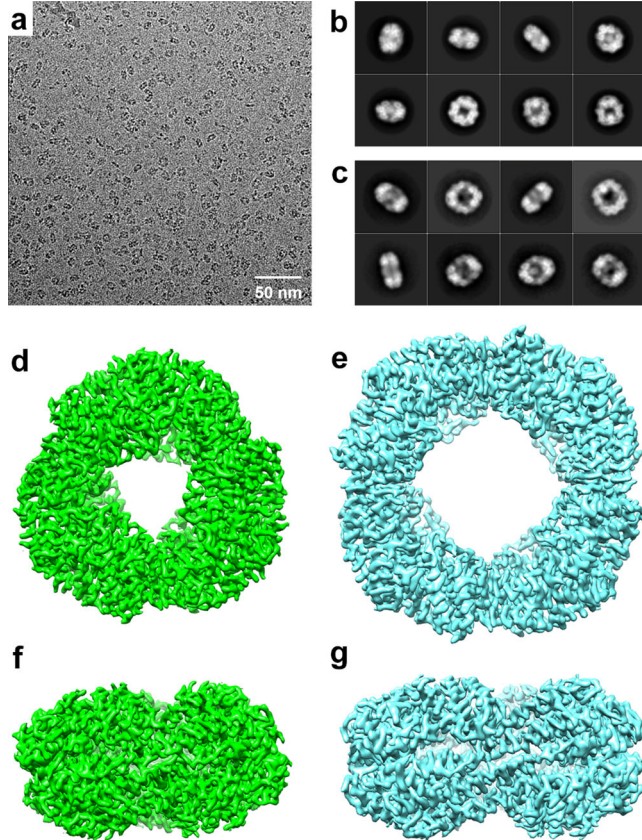

**Fig. 3 Cryo-EM analyses of *Tl*CPC-6 and *Tl*CPC-8.** Representative motion-corrected electron micrograph of *Tl*CPCs (**a**). Typical reference-free 2D class averages from single-particle images of *Tl*CPC-6 (**b**) and *Tl*CPC-8 (**c**). Top and side views of cryo-EM density maps of *Tl*CPC-6 at 3.06 Å resolution (**d** and **f**) and *Tl*CPC-8 at 3.71 Å resolution (**e** and **g**). Contour levels of *Tl*CPC-6 and *Tl*CPC-8 are shown at 0.06 and 0.04, respectively.

**Table 2 Cryo-EM data collection, refinement, and validation statistics.**

| (EMDB/PDB ID) | *Tl*CPC-6 (EMD-31090/7EH8) | *Tl*CPC-8 (EMD-31089/7EH7) |
|---|---|---|
| ***Data collection and processing*** | | |
| Microscope | Talos Arctica | Talos Arctica |
| Voltage (kV) | 200 | 200 |
| Detector | Falcon 3EC | Falcon 3EC |
| Magnification | 120,000 | 120,000 |
| Pixel size (Å) | 0.88 | 0.88 |
| Automation software | EPU | EPU |
| Total exposure (e⁻/Å²) | 50 | 50 |
| Exposure rate (e⁻/Å² fraction) | 1.02 | 1.02 |
| Number of frames | 49 | 49 |
| Defocus range (μm) | −1, −1.5, −2, −2.5 | −1, −1.5, −2, −2.5 |
| Number of collected micrograph | 2,036 | 2,036 |
| Number of particles for Class2D | 283,980 | 129,653 |
| Number of particles for Class3D | 72,655 | 23,643 |
| Number of particles for Refine3D | 28,120 | 10,402 |
| Symmetry imposed | $D_3$ | $D_4$ |
| Map resolution (Å) | 3.06 | 3.71 |
| FSC threshold | 0.143 | 0.143 |
| Map resolution range (Å) | 2.91–3.38 | 3.47–4.49 |
| ***Refinement*** | | |
| Refinement programs | PHENIX/Coot | PHENIX/Coot |
| Initial model used (PDB code) | 7EFW | 7EFV |
| Map-to-model resolution (Å) | 3.05 | 3.75 |
| FSC threshold | 0.5 | 0.5 |
| Model resolution range (Å) | 2.91–3.38 | 3.47–4.49 |
| ***Model composition*** | | |
| Non-hydrogen atoms | 15773 | 20904 |
| Protein residues | 2004 | 2656 |
| Ligands | 18 | 24 |
| ***B factors (Å²)*** | | |
| Protein | 38.8 | 51.9 |
| Ligand | 39.4 | 51.2 |
| ***Map-model CC*** | | |
| CC (mask) | 0.90 | 0.86 |
| CC (box) | 0.77 | 0.80 |
| CC (peaks) | 0.75 | 0.75 |
| CC (volume) | 0.85 | 0.83 |
| ***R.m.s. deviations*** | | |
| Bond lengths (Å) | 0.003 | 0.004 |
| Bond angles (°) | 1.051 | 1.298 |
| ***Validation*** | | |
| MolProbity score | 1.3 | 1.7 |
| Clash score | 5.5 | 8.7 |
| Poor rotamer (%) | 0.79 | 1.69 |
| ***Ramachandran plot*** | | |
| Favored (%) | 98.2 | 97.8 |
| Allowed (%) | 1.8 | 2.2 |
| Disallowed (%) | 0 | 0 |

potassium phosphate buffer solution of *Tl*CPC was applied to a holey carbon grid and then flash-frozen to prepare a cryo-grid for structural determination (Supplementary Note 1). Cryo-EM micrographs and selected reference-free 2D class averages clearly showed two types of ring-shaped oligomeric assemblies in frozen-hydrated specimens as expected (Fig. 3a). Accordingly, each assembly could be subjected to single particle analysis (Fig. 3b, c; Supplementary Figs. 15–18), which resulted in reconstructions of hexameric $(\alpha\beta)_6$ and octameric $(\alpha\beta)_8$ states with overall resolutions of 3.06 and 3.71 Å, respectively (Fig. 3d–g; Table 2). The EM reconstruction models were essentially isostructural to the crystal models (*Tl*CPC-6: rmsd value of 1.14 Å over 1,923 Cα atoms; *Tl*CPC-8: rmsd value of 1.39 Å over 2,565 Cα atoms) (Supplementary Fig. 19), supporting the coexistence of conventional hexameric $(\alpha\beta)_6$ and novel octameric $(\alpha\beta)_8$ states in the frozen solution. These results strongly indicated that the concentrated solution of the native *Tl*CPC contained hexamers and octamers. The cryo-EM model of *Tl*CPC-6 was essentially isostructural to the cryo-EM models of PCs in PBSs from *G. pacifica* (*Gp*PC; PDB ID: 5Y6P) and *P. purpureum* (*Pp*PC; PDB ID: 6KGX) (*Gp*PC: rmsd value of 1.06 Å over 1,887 Cα atoms; *Pp*PC: rmsd value of 0.96 Å over 1,932 Cα atoms) (Supplementary Fig. 20). Although low-resolution EM structures from CPC and high-resolution cryo-EM structures from PCs in PBSs have been reported, the cryo-EM models of *Tl*CPC-6 and *Tl*CPC-8 are the first examples of high-resolution cryo-EM structures from CPC.

The local resolution maps of *Tl*CPC-6 and *Tl*CPC-8 showed relatively low resolutions at helices $C_\alpha$, $D_\alpha$, $A_\beta$, $B_\beta$, $C_\beta$, $D_\beta$, $F_\beta$, and $G_\beta$ and in the loops $D/E_\alpha$, $D/E_\beta$, $F/G_\beta$, and $H/I_\beta$ (Supplementary Fig. 21). These helices and loops also possessed relatively high B-factors in the crystal models of *Tl*CPC-6 and *Tl*CPC-8 (Supplementary Fig. 22). Therefore, the structural fluctuations of these regions are relatively large regardless of the effects of

crystal packings (X-ray) and damages induced by freezing processes (cryo-EM).

**Geometries of *Tl*CPC-6 and *Tl*CPC-8.** Interestingly, the X-ray crystallographic, cryo-EM, and MALDI-TOF mass analyses of *Tl*CPC clearly indicate that different oligomeric states of *Tl*CPC-6 and *Tl*CPC-8 were assembled from the same monomeric ($\alpha\beta$) units as described above. Investigating the structures of homologous proteins with different oligomeric states is crucial to elucidating their functions in metabolism as well as the mechanisms of protein assembly and evolution[46,47]. In general, quaternary structures are conserved among proteins with the same or very high sequence identities because the misassembly of oligomers in vivo is implicated in physiological disorders[46,48]. However, especially in closed circular oligomers with cyclic ($C_n$) or dihedral ($D_n$) symmetry ($n > 2$), it has been reported that different oligomeric states from identical subunits, including protective antigens of toxins ($n = 7$, 8, 38, and 44)[49,50], portal proteins ($n = 12$ and 13)[51], flagellar motors ($n = 32–36$)[52], and *trp* RNA-binding attenuation protein ($n = 11$ and 12)[53], play important roles in various biological regulations to control diameter and curvature of ring-shaped proteins. These studies have shown that slight conformational changes in secondary and tertiary structures enable the control of rotation angles between adjacent subunits [$\Delta\theta$, ideally equal to $(360/n–360/m)°$ for oligomers with $n$- and $m$-fold rotation axes ($n < m$)][53]; in contrast, only theoretical studies have been conducted in oligomers with a small number of $n$-fold rotation axes ($n = 3$ and 4) because of the requirement of large structural changes ($\Delta\theta = 30°$) (Supplementary Fig. 23)[54,55]. In this context, *Tl*CPC-6 and *Tl*CPC-8 with their respective $D_3$ and $D_4$ symmetries are the first examples of identical subunits being assembled into different cyclic oligomers with 3- and 4-fold rotation axes.

**Analytical ultracentrifugation of *Tl*CPC.** To date, size-exclusion chromatography and analytical ultracentrifugation have mainly been utilized to analyze the oligomeric states of CPC in solutions. Thoren *et al.* reported that a monomeric ($\alpha\beta$) state in solution could be obtained by size-exclusion chromatography, whereas a homogeneous elution of trimeric ($\alpha\beta$)$_3$ or hexameric ($\alpha\beta$)$_6$ states could not be obtained partly due to the aforementioned trimer–hexamer equilibrium of diluted CPC samples in the column[29,32]. On the other hand, analytical ultracentrifugation has long been a powerful tool by which to estimate the oligomeric states of CPCs in solutions. Berns and MacColl previously determined the sedimentation coefficients of CPCs under various conditions and established a widely accepted interpretation as follows: monomeric ($\alpha\beta$), trimeric ($\alpha\beta$)$_3$, and hexameric ($\alpha\beta$)$_6$ species were observed at sedimentation coefficient distributions of 3, 7, and 11 S, respectively[31].

Despite this established consensus, there has been an exceptional observation of dimeric ($\alpha\beta$)$_2$ states: in a report by Neufeld and Riggs, analytical ultracentrifugation of the CPC from *S. elongatus* PCC 7942 (*Se*CPC, previously known as *Anacystis nidulans*) showed monomer–dimer–hexamer equilibrium in solutions[56]. Although X-ray crystallographic analysis later revealed that the *Se*CPC was a hexameric state in crystals[23], dimeric ($\alpha\beta$)$_2$ states in solutions could be obtained depending on the specific microorganisms or isolation methods.

To investigate the solution states of diluted *Tl*CPC, analytical ultracentrifugation was performed in the present study. Analytical ultracentrifugation of *Tl*CPC (0.92 mg/mL) in potassium phosphate buffer (pH 7.0, 10 mM) gave a major sedimentation coefficient distribution at 5.24 S ($s_{20,w} = 5.56$ S) and minor sedimentation coefficient distributions at 3.15 S ($s_{20,w} = 3.33$ S), 7.97 S ($s_{20,w} = 8.45$ S), and 11.32 S ($s_{20,w} = 12.00$ S) (Supplementary Fig. 24). According

to the reported *s* values, observed species distributed at 3.15, 5.24, and 11.32 S were assignable to monomeric ($\alpha\beta$), dimeric ($\alpha\beta$)$_2$, and hexameric ($\alpha\beta$)$_6$ states, respectively, whereas the species distributed at 7.97 S is observed for the first time here. The molecular weights of these species were calculated using the Svedberg equation as follows: 37.1 kDa (for 3.15 S), 79.8 kDa (for 5.24 S), 149.5 kDa (for 7.97 S), and 253.1 kDa (for 11.32 S). The obtained values of 37.1 and 79.8 kDa were in agreement with the calculated masses of the monomer (37.3 kDa) and dimer (74.7 kDa), respectively, whereas the value of 253.1 kDa was slightly larger than expected (hexamer: 224.1 kDa). Interestingly, the obtained weight of the previously unobserved distribution at 7.97 S (149.5 kDa) was similar to the calculated mass of the tetramer (149.4 kDa). It should be noted that Iso *et al.* previously reported the presence of tetramers in a solution[57]. Overall, these results indicate that the unusual dimeric ($\alpha\beta$)$_2$ state in the solution was a key intermediate structure in the assembly of the newly observed octameric ($\alpha\beta$)$_8$ state.

**Assembly mechanism of *Tl*CPC.** Based on the results reported thus far, an assembly mechanism for *Tl*CPC is proposed. The crystal and cryo-EM models of *Tl*CPC-6 and *Tl*CPC-8 clearly showed slight structural differences of monomer–monomer interfaces (helices G$_\alpha$, C$_\beta$, and E$_\beta$ and loop D/E$_\beta$) and $\alpha$–$\beta$ subunit interfaces (helices A$_\alpha$, B$_\alpha$, E$_\alpha$, F$_\alpha$, A$_\beta$, B$_\beta$, C$_\beta$, and E$_\beta$) between *Tl*CPC-6 and *Tl*CPC-8; The superimposition of *Tl*CPC-6 and *Tl*CPC-8 monomers showed that slight structural differences existed at helices A$_\alpha$, B$_\alpha$, A$_\beta$, B$_\beta$, C$_\beta$, and D$_\beta$ and loops A/B$_\alpha$, D/E$_\alpha$, A/B$_\beta$, and D/E$_\beta$ (Fig. 1e–h; Supplementary Fig. 25). A possible basis for these structural changes were high B-factors and low local resolutions observed by the X-ray crystallographic and cryo-EM analyses, respectively (Supplementary Figs. 21 and 22), suggesting the relatively flexible protein backbones at these interfaces. Therefore, extended cyclic structures were possible because the flexible interfaces acted as "hinges", which enabled to control of the curvatures of protein rings although *Tl*CPC-6 and *Tl*CPC-8 possessed the same amino acid sequences. In addition, Adir *et al.* reported that the hydrogen bond network between $\alpha$Asp28, $\beta$Asn35, and $\beta$PCB153 was critical to form hexameric structure in *Tv*CPC[16,23], while *Tl*CPC possesses $\beta$Ser35 that is rarely observed in the previously reported CPC crystals: (1) the direct hydrogen bond between $\beta$Asn35–$\beta$153PCB was observed in *Tv*CPC, whereas the indirect hydrogen bond between $\beta$Ser35–$\beta$153PCB via a water molecule was observed in *Tl*CPC, and (2) one of the two indirect hydrogen bonds between $\beta$Asn35–$\alpha$Asp28 in *Tv*CPC was not observed in *Tl*CPC-6, suggesting the destabilization of hexameric structures in *Tl*CPC (Supplementary Fig. 26a). Moreover, the indirect hydrogen bond between $\alpha$Lys32–$\beta$153PCB in *Tv*CPC is important to stabilize ($\alpha\beta$) monomer, whereas $\alpha$Glu32 in *Tl*CPC rather stabilizes a dimeric structure ($\alpha\beta$)$_2$ by interacting with the neighboring $\beta$ subunit in the dimeric interface (Supplementary Figs. 26b and 27a). These unique residues might be responsible for unusual octamer assembly.

The PISA program calculated the solvation free energies of oligomeric ($\alpha\beta$)$_2$, ($\alpha\beta$)$_3$, ($\alpha\beta$)$_4$, ($\alpha\beta$)$_6$, and ($\alpha\beta$)$_8$ states ($\Delta G_{int}$) as −87.4, −162.3, −204.1, −352.3, and −420.3 kcal/mol, respectively. These values increased as the accessible surface area of each oligomer increased, where the structure of the dimer was assumed to be one-third of the hexamer [contact% = buried surface area / (accessible surface area + buried surface area) = 26%] because a dimer in two-thirds of a trimer is expected to be unstable in solution (contact% = 4%) (Supplementary Fig. 27). The dissociation free energies of oligomeric ($\alpha\beta$)$_2$, ($\alpha\beta$)$_3$, ($\alpha\beta$)$_4$, ($\alpha\beta$)$_6$, and ($\alpha\beta$)$_8$ states into monomeric ($\alpha\beta$) states ($\Delta G^0_{diss}$) were also calculated as 4.8, 20.9, 16.0, 54.6, and 20.9 kcal/mol, respectively, clearly indicating that hexamers were thermodynamically more stable than octamers. These results also revealed that trimers were

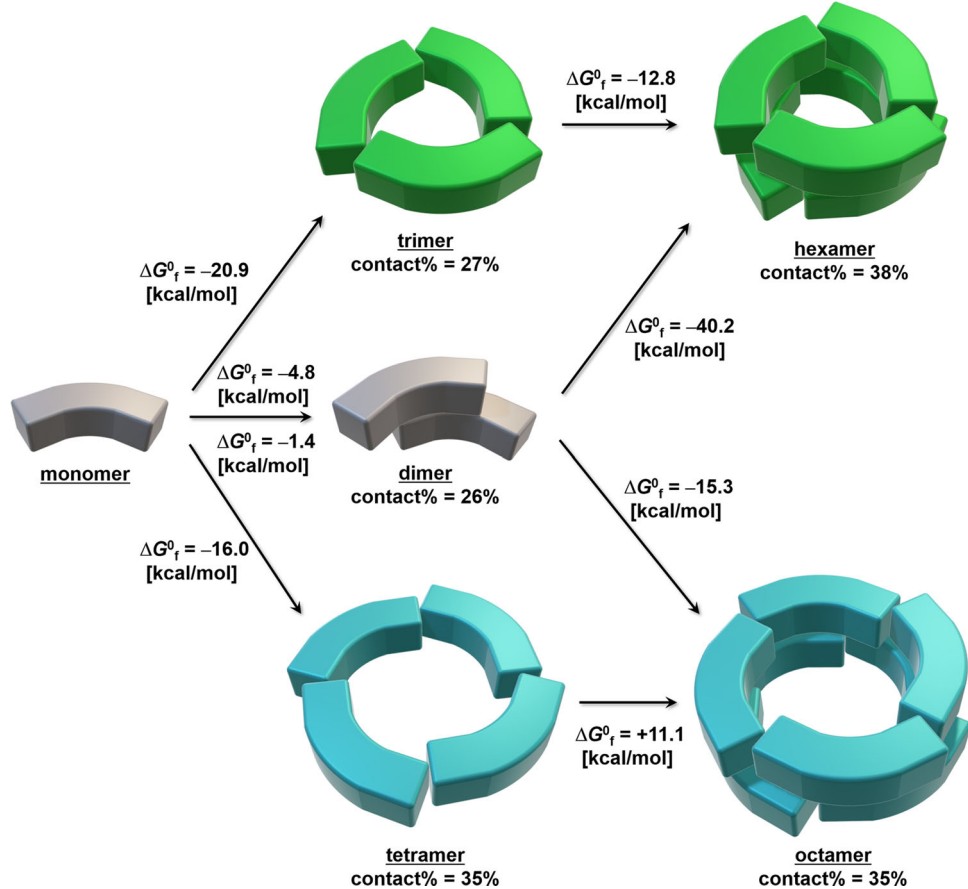

**Fig. 4 Proposed oligomerization mechanism in *Tl*CPC.** Monomeric (αβ) structures are represented by bended cuboids. Contact% was calculated by the PISA program based on the monomeric (αβ) units. $\Delta G^0_f$ values were calculated from $\Delta G^0_{diss}$ values, where the $\Delta G^0_f$ values of –4.8 and –1.4 kcal/mol in the dimer were calculated using the crystal models of *Tl*CPC-6 and *Tl*CPC-8, respectively.

thermodynamically more stable than tetramers, which is consistent with findings that all previously observed CPCs in crystals have been trimers or hexamers. Therefore, the formation of the hexamer from monomers is thought to be the major assembly route for CPCs.

By utilizing the $\Delta G^0_{diss}$ values, the Gibbs free energies of formation ($\Delta G^0_f$) between oligomers were calculated to further investigate the assembly process (Fig. 4). Interestingly, the $\Delta G^0_f$ value from tetramers into an octamer was positive (11.1 kcal/mol), which indicated that octamers are difficult to assemble from tetramers, whereas the $\Delta G^0_f$ value from dimers into an octamer was negative. These results supported the proposed monomer–dimer–octamer assembly process as a plausible process by which *Tl*CPC-8 is formed. Therefore, the formation of unusual dimeric (αβ)₂ states may be kinetically more favorable than the formation of trimers or tetramers in *Tl*CPC, which supports the conclusion that dimers are key intermediate structures in the assembly of the newly observed octameric (αβ)₈ state in *Tl*CPC.

In summary, an unusual octameric (αβ)₈ state in *Tl*CPC was presumably formed by (1) adjusting the dihedral angles of flexible monomeric (αβ) units, (2) assembling monomers into kinetically favored dimeric (αβ)₂ states, and (3) assembling four dimers into a cyclic structure. Although the key parameter for switching the hexameric and octameric states of *Tl*CPC remains under investigation, one possible factor is that crystallization of the solution, freezing of the solution, and analysis of the solution by ultracentrifugation were performed soon after isolating native CPCs from O-77 at 4–20 °C and kinetically controlled. These in vitro results expand the current perspective of the PBP assembly process in vivo.

## Conclusion

In this study, a previously unobserved octameric state of CPC that differed from the conventional hexameric state was discovered for the first time. The existence of non-conventional octameric CPC was confirmed by X-ray crystallographic analysis and the first high-resolution cryo-EM analysis of the native CPC from O-77. Although the monomers in both hexameric and octameric states were essentially identical, the structural analysis indicated that slight conformation changes in potentially flexible interfaces and unique amino acid residues enabled the formation of different oligomeric states from the same primary structures. The unusual dimeric states observed via analytical ultracentrifugation were presumably key intermediate structures in the assembly of the octamer in *Tl*CPC; however, the proposed monomer–dimer–octamer assembly process might be observed in other CPCs because *Tl*CPC shares relatively high sequence identities (α: 68–86%; β: 65–87%) with the previously reported hexameric CPCs in crystals. Since the assembly of identical subunits into symmetrically different oligomeric states is rare and interesting in biology, the experimental results reported in this study are important for understanding not only the assembly process of oligomers but also the evolution process of proteins. Although the biological significance of the octameric structure remains to be clarified, we will focus on it in future studies. We believe that this study provides new insights into the assembly processes of CPCs and PBSs both in vivo and in vitro and sheds new light on the mechanism of energy transfer in the light-harvesting complexes of cyanobacteria.

## Methods

**Chemicals**. Potassium hydroxide, potassium dihydrogen phosphate, sodium chloride, 2-amino-2-hydroxymethyl-1,3-propanediol (Tris), and hydrochloric acid were purchased from FUJIFILM Wako Pure Chemical Corporation and used as received.

**Purification of CPC**. Cells of O-77 were grown according to a previously reported procedure[36]. The cells were harvested and homogenized with 10-mM Tris/HCl buffer at pH 8.0 (buffer A) using NZ-1000 (EYELA, Japan) and then disrupted in an ice bath by sonication three times (2-min sonication at 30 W with a 2-min break) using an Ultrasonic Disruptor UD-200 (TOMY SEIKO, Japan). Cell debris and unbroken cells were removed by centrifugation (5000 $g$, 20 min, 277 K) using a Himac CR20GII (HITACHI, Japan), and the resulting supernatant was centrifuged at 100,000 $g$ for 1 h using an Optima L-90K (Beckman Coulter, USA). The supernatant of soluble cell extracts was then loaded onto a DEAE Sepharose fast flow column (XK 26/20; GE Healthcare Life Sciences, UK) preequilibrated with buffer A by washing with 500 mL of the same buffer at a flow rate of 10 mL min$^{-1}$. The $Tl$CPC-containing solution was eluted at 0.15–0.24 M NaCl with 10–19 mS cm$^{-1}$ ionic conductivity using buffer A alone and buffer A containing 1.0 M NaCl (buffer B) as eluents. Fractions containing $Tl$CPC were combined and diluted threefold with buffer A and the resulting solution was loaded onto a Q Sepharose high-performance column (HR 16/10; GE Healthcare Life Sciences) preequilibrated with buffer A by washing with 150 mL of the same buffer at a flow rate of 4 mL min$^{-1}$. Blue $Tl$CPC-containing solution was eluted at 0.15–0.20 M NaCl with 13–17 mS cm$^{-1}$ ionic conductivity using buffers A and B. The fractions containing $Tl$CPC were combined and concentrated using Amicon Ultra-15 50 kDa (Merck, Germany). The resulting concentrated $Tl$CPC was purified using a Superdex 200 prep grade column (HR 16/50; GE Healthcare Life Sciences) and a potassium phosphate buffer solution (pH 7.0, 10 mM) as an eluent. The concentration of the purified $Tl$CPC solution was determined by the following equation[58]:

$$\text{Concentration of CPC} \, (\text{mg/mL}) = [A_{615} - 0.474(A_{652})]/5.34,$$

where $A_{615}$ and $A_{652}$ are the absorption intensities at 615 and 652 nm, respectively, which were obtained by measuring the UV–vis spectrum.

**Spectral analysis**. Circular dichroism was measured using Chirascan (Applied Photophysics, UK) with a 0.1-cm quartz cell at 20 °C. The UV–vis spectrum was measured on JASCO V-670 (JASCO, Japan) with a 1-cm quartz cell. The fluorescence spectrum with an excitation wavelength of 436 nm was measured using a FluoroMax-4 Spectrofluorometer (HORIBA Jobin Yvon, NJ).

MALDI-TOF mass spectrometric analysis of CPC solutions were performed using a microflex LT (Bruker Daltonics, Billerica, MA, USA). The saturated sinapinic acid solution in ethanol was deposited onto a ground steel MALDI target plate to form a matrix layer. A mixture of CPC and sinapinic acid in a mixed solvent of acetonitrile and 0.1% trifluoroacetic acid (3/7, v/v) was then dropped on the matrix layer and dried at room temperature (ca. 25 °C). The resulting sample was deposited onto a ground steel MALDI target plate and dried at room temperature (ca. 25 °C).

Analytical ultracentrifugation was performed on an Optima AUC analytical ultracentrifuge (Beckman Coulter) in a four-hole An60Ti rotor at 20 °C. Samples (0.92 mg/mL) in potassium phosphate buffer solution (pH 7.0, 10 mM) were set in aluminum double-sector centerpieces with quartz windows. Concentration profiles were monitored by absorbance at 277 nm because the absorbance at 615 nm is saturated at 0.92 mg/mL of $Tl$CPC. Kao *et al.* reported that scans of the same sample measured at 620 and 278 nm yielded the same result[59]. The rotor speeds were set at 60,000 rpm. Without intervals between scans, scans were continually recorded until all of the solutes were sedimented at the bottom. Sedimentation velocity data were analyzed by SEDFIT[60] to obtain the distribution of sedimentation coefficients using the continuous $c(s)$ distribution model based on the Lamm equation as follows:

$$a(r,t) \cong \int_{s_{\min}}^{s_{\max}} c(s)\chi(s, D(s), r, t)ds,$$

where $r$ is the radius from the center of rotation, $s$ is the sedimentation coefficient, $t$ is the time in seconds, and $D$ is the diffusion coefficient. The molecular weight of the corresponding sedimentation coefficient was also calculated by SEDFIT using the Svedberg equation as follows:

$$\frac{s}{D} = \frac{M(1 - \bar{v}\rho)}{RT},$$

where $M$ is the molecular weight, $\bar{v}$ is the partial specific volume, $\rho$ is the density of the solution, $R$ is a gas constant, and $T$ is the temperature. The solution density and viscosity of the buffer solution were calculated with SEDNTERP[61] and set at 1.00980 g/mL and 0.01029 P, respectively. The partial specific volume was also calculated with SEDNTERP using the amino acid sequences of the α and β subunits in $Tl$CPC; it was set at 0.73149 mL/g. The frictional ratio was initially set at 1.2 for solving the Lamm equation and then set as a variable for fitting the data. The best-fit frictional ratio was 1.195 with a rmsd of <0.01.

**Crystallization, data collection, and structural determination and refinement**. The purified $Tl$CPC was crystallized using the sitting-drop vapor diffusion method at 20 °C. Sitting drops contained 200 nL of protein solution mixed with 200 nL of reservoir solution. Two types of crystals ($Tl$CPC-6 and $Tl$CPC-8) were obtained (Supplementary Table 1 and Supplementary Fig. 12). For $Tl$CPC-6 crystals (rhombus or rod), the reservoir condition contained 8% Tacsimate (pH 5.0) and 20% polyethylene glycol 3,350. Prior to data collection, crystals were transferred to a cryoprotectant solution containing polyethylene glycol 3,350 and then flash-cooled to –180 °C. For $Tl$CPC-8 crystals (cube), the reservoir condition contained 1.6-M sodium chloride, 8% polyethylene glycol 6,000, and 20% glycerol, i.e., a cryoprotectant solution. The crystals were then flash-cooled to –180 °C. The cubic crystals of octamers determined by the X-ray crystallographic analysis were also obtained at 42 °C which is the growth temperature of O-77 in this study.

X-ray diffraction data were collected at beamline BL45XU at Spring-8 (Hyogo, Japan) and 100 K with a wavelength of 1.000 Å. These data were processed using the ZOO system[62,63]. Phases were determined by molecular replacement using the program Phaser and search models of monomers of $Tv$CPC (PDB ID: 1I7Y)[16]. Models were built using the program Coot[64], and the program Phenix.refine[65] was used for refinement. The crystal models displayed good geometry when analyzed by MolProbity[66]. In the Ramachandran plots, 98.3% and 1.7% of the residues constituting $Tl$CPC-6 were in the most favored and allowed regions, whereas 98.0% and 2.0% of the residues constituting $Tl$CPC-8 were in the most favored and allowed regions, respectively.

**Cryo-EM sample preparation, data collection, and data processing**. For cryo-grid preparation, 3 μL of $Tl$CPC samples (6.7 and 10 mg/mL) in potassium phosphate buffer solution (pH 7.0, 10 mM) were applied onto a holey carbon grid (Quantifoil, Cu, R1.2/1.3, 300 mesh). The grid was rendered hydrophilic by a 30 s glow-discharge in the air (11 mA current) with PIB-10 (Vacuum Device Inc., Ibaraki, Japan). The grid was then blotted for 5 sec (blot force 15) at 18 °C and 100% humidity before being flash-frozen in liquid ethane using Vitrobot Mark IV (Thermo Fisher Scientific, Waltham, MA, USA). For automated data collection, 2,036 micrographs were acquired on a Talos Arctica (Thermo Fisher Scientific) microscope operating at 200 kV in nanoprobe mode and using EPU software. The movie micrographs were collected on a 4k × 4k using a Falcon 3EC direct electron detector (electron counting mode) at a nominal magnification of 120,000 (0.88 Å/pixel). Forty-nine movie fractions were recorded at an exposure of 1.02 electrons per Å$^2$ per fraction, which corresponded to a total exposure of 50 e$^-$/Å$^2$. The defocus steps used were –1.0, –1.5, –2.0, and –2.5 μm. The movies were processed by MotionCor2[67], Gctf[68], and RELION3.0[69]. The cryo-EM models were visualized by UCSF chimera[70]. See Supplementary Note 1 and Supplementary Figs. 15–18 for the details of the cryo-EM data processing.

**Reporting summary**. Further information on research design is available in the Nature Research Reporting Summary linked to this article.

## Data availability

Crystal structures are deposited at the Protein Data Bank with accession codes 7EFW ($Tl$CPC-6) and 7EFV ($Tl$CPC-8). Cryo-EM maps are deposited in the Electron Microscopy Data Bank under accession codes EMD-31090 ($Tl$CPC-6) and EMD-31089 ($Tl$CPC-8). Structure coordinates related to the cryo-EM maps are deposited at the Protein Data Bank with accession codes 7EH8 ($Tl$CPC-6) and 7EH7 ($Tl$CPC-8). Any remaining information can be obtained from the corresponding authors upon reasonable request.

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

## Acknowledgements
*Thermoleptolyngbya* sp. O-77 was named after Saori Ogo, whose assistance was invaluable to the hunt for a new thermophilic cyanobacterium. This paper is dedicated to her memory. This work was supported in part by JST CREST Grant Number JPMJCR18R2 (S.O.), JSPS KAKENHI Grant Numbers JP18H02091 (K.-S.Y) and JP18J00191 (T. Minato), MEXT Leading Initiative for Excellent Young Researchers Grant Number JPMXS0320200400 (T. Minato), JSPS Core-to-Core Program, the World Premier International Research Center Initiative (WPI), Japan, and the Platform Project for Supporting Drug Discovery and Life Science Research (Basis for Supporting Innovative Drug Discovery and Life Science Research (BINDS)) from the Japan Agency for Medical Research and Development (AMED) under Grant Number JP21am0101071 (supporting no. 2493). T. Minato was supported by the JSPS through a Research Fellowship for Young Scientists. We thank the staff members of the beamline BL45XU facilities at SPring-8 for their help with X-ray data collections. The cryo-EM data were collected at the Cryo-EM facility in KEK (Ibaraki, Japan).

## Author contributions
T. Minato, T.T., Y.K., S.O. and K.-S.Y. devised the study. T. Minato, N.K.H., K.Y. and K.-S.Y. isolated and characterized the protein. T.T. and Y.K. performed X-ray crystallographic and cryo-EM structural analyses. N.A., M.K., M.A., T. Moriya and T.S. performed cryo-EM data collection, processing, and cryo-EM structural analysis. T. Minato and T.T. wrote the paper.

## Competing interests
The authors declare no competing interest.
