## [Peer Review File · Communications Biology]

Reviewers' comments:

Reviewer #1 (Remarks to the Author):

The manuscript of Minato et al. describes an unusual structure for a cyanobacterial phycocyanin—a unique octomeric structure for phycocyanin that apparently coexists in solution with the more common place heteromeric structure. The authors present structures based upon both X-ray crystallographic studies as well as cryo-electron microscopy. In general, the manuscript is concisely and clearly written, and presents the results in an engaging and well-organized manner. Only a few minor comments for the authors follow.

1. The authors refer to this protein as CPC, in which the first C means “cyanobacterial.” It really is unnecessary to use this letter and throughout the protein could be referred to more simply as PC, phycocyanin.
2. Line 43. Three dimensional objects such as phycocyanin hexamers are not well-described as two-dimensional objects (e.g., a circle). Phycocyanin is shaped like a bagel; in other words, it is a toroid.
3. Line 43-44. Delete ...although...determined.
4. Line 51-52. It is difficult to say if there was/is a belief that phycocyanin only assembles into hexamers. It was well-established that monomers and trimers also occur, and some phycocyanins have been crystallized in the trimeric assembly form. Given that no oligomers other than monomers ($\alpha\beta$)₁, dimers ($\alpha\beta$)₂, trimers ($\alpha\beta$)₃, and hexamers ($\alpha\beta$)₆ had been described, this is not so much a “belief” as it was “these are the only known assembly forms.” The authors do not yet know the significance of the octomers ($\alpha\beta$)₈ detected, only that they can occur.
5. Line 63. Stacked, not pile. Or, stacks of...
6. General. The authors do not state the growth temperature optimum for *Thermoleptolyngbya* sp., but they are working at room temperature, which is far from optimal for a thermophile. Is the ($\alpha\beta$)₈ octomer an artifact of the lower temperature? The authors show that the ($\alpha\beta$)₆ hexamer is much more stable, and perhaps ($\alpha\beta$)₈ octomers are only stable at room temperature and not at the temperature at which this organism normally grows.
7. Line 106. Dissociated, not dissolved
8. Line 115. The authors appear to have the genome sequence for this organism, so they must know whether the genome encodes single genes for *cpcA* and *cpcB*, encoding the alpha and beta subunits for phycocyanin. This should be stated in the text. However, even if there are single genes, this does not mean that the two phycocyanins are identical, because they could be differently modified posttranslationally. Did the authors perform mass spectrometry on the two phycocyanins to demonstrate that the masses of the subunits in the two assembly forms were identical? More details about the identity of the two sets of subunits should be provided in the manuscript. Also, the authors do not compare the absorbance spectra, CD spectra, etc. for the two types of PC. These data should also be shown.
9. Line 120-122. The authors should cite Chang et al. (Cell Research, 25, 726-737 (2015)), who studied the structure of the pentacylindrical PBS of *Nostoc* (*Anabaena*) sp. PCC 7120.
10. Sweet et al. (J. Biol. Chem. 252, 8258-8260, 1977) showed that crystals of PC from *Anabaena variabilis* produced crystals of space group P6 with a dimer ($\alpha\beta$)₂ as the asymmetric unit. This could be relevant to the discussion here and should be cited.
11. Concerning assembly in vivo, aside from the effects of temperature, which the authors did not explore here, it is also possible that interactions with linker proteins in cells could modify the dynamics of the assembly process. Presumably, most of the PC ($\alpha\beta$)₆ hexamers would bind linkers and be inserted into PBS. Alternatively, there might be alternative linkers that could make use of the octomeric form of PC (e.g., assemblies together with another linker such as CpcL). Does the genome encode a CpcL-type linker polypeptide?

Reviewer #2 (Remarks to the Author):

This work finds and shows the unique oligomerization of TICPC. It is interesting and significant to light-harvesting of photosynthesis. The unique octamer of TICPC may be promising for bioengineering light-harvesting units and photosensory units.

This reviewer has the following concerns:

Line 127-128, How do the authors explain the discrepancy between 22.5 and 17.9 kDa for beta-TICPC? May be it important to explain the unique oligomerization of TICPC, when the discrepancy corresponds to a (post-translational) modification in apo-protein. Did the authors measure the mass of beta-TICPC using Mass Spectrometry (MS)? When not, why? When yes, what are the MS results? The mass difference between alpha and beta-TICPC in SDS-PAGE (Supplementary Fig. 1) is surprisingly great, so the MS results are key to elucidate the mass difference, which might be the molecular basis of the unexpected oligomerization of TICPC.

The authors' data show that the resolution of helices A and B, which are responsible to oligomerization, is lower (Supplementary Fig. 18). The authors should compare the sequences of the N-termini (1-30 aa) among CPCs, and compare the segments that have the higher B-factor (Supplementary Fig. 19) in alpha- and beta-TICPC with the related segments of other CPCs including PCs from cryptophytes. PCs from cryptophytes that are homologous to CPCs have also unique oligomerization. The N-termini of phycobiliproteins are responsible to oligomerization. The segments of higher B-factor might make the difference in oligomerization. In Supplementary Fig. 4, the helix B of alpha-TICPC contains E instead of K/R in other most alpha-CPCs. Does the exchange of acidic/basic amino acids make sense in oligomerization? Comprehensive analyses combining sequences with structures should shed some light on this unique oligomerization.

On description and discussion of CPC biosynthesis, the cited reference(s) are too old (e.g. 1998 in lines 92-94). According to the works of CPC biosynthesis since 2000, the CPC biosynthesis undergoes PCB-chromophorylation and methylation of subunits (the two steps are hardly involved in oligomerization), and then oligomerization. Namely the two post-translational modifications are the prerequisites for the (correct) oligomerization of CPCs.

Reviewer #3 (Remarks to the Author):

This study shows structurally the apparently the same PC subunits can oligomerise into two separate forms, hexamers and octamers. This is novel as so far only hexamers have been characterised. However, in the absence of other data this observation is just a curiosity. the major question that is not addressed is does the octamer have any functional relevance. Without that this paper lacks significance.

there are also a range of question that the data presented raise that need to be answered. The SDS gel of the complex is very underlapped. This means that the evidence that the sample is pure is very poor. The authors also say that the higher molecule weight subunit runs at too high a mobility on the gel judged by its actual mass. Their observation is just left hanging. So why is this true? What proportion of the PC sample assembles into hexamers and what into octamers? If the hexamers are monomerised to they still get a mixture of oligomeric states when the sample is allowed to reoligomerise? Also vice versa with the octamers? The regions of the suture that change upon formation of the octamers rather than the hexamers needs a much fuller discussion. If they look at the sequence of their PC relative those that only form hexamers can any structural motifs be found that could suggest why this occurs? In other words when more gene sequences of PCs from new organisms are found can they use that information to predict those sequences that could support octamerisation?

Also in vivo, linker polypeptides are important for formation of the phycobilisome rods. Is octamerisation affected by the presence of linkers? if so them maybe the octamers can only form in the absence of linkers?

Referees' comments (COMMSBIO-21-1096-T)

Referee 1:

Reviewer #1 (Remarks to the Author):

The manuscript of Minato et al. describes an unusual structure for a cyanobacterial phycocyanin—a unique octomeric structure for phycocyanin that apparently coexists in solution with the more common place heteromeric structure. The authors present structures based upon both X-ray crystallographic studies as well as cryo-electron microscopy. In general, the manuscript is concisely and clearly written, and presents the results in an engaging and well-organized manner. Only a few minor comments for the authors follow.

1. The authors refer to this protein as CPC, in which the first C means “cyanobacterial.” It really is unnecessary to use this letter and throughout the protein could be referred to more simply as PC, phycocyanin.
2. Line 43. Three dimensional objects such as phycocyanin hexamers are not well-described as two-dimensional objects (e.g., a circle). Phycocyanin is shaped like a bagel; in other words, it is a toroid.
3. Line 43-44. Delete ...although...determined.
4. Line 51-52. It is difficult to say if there was/is a belief that phycocyanin only assembles into hexamers. It was well-established that monomers and trimers also occur, and some phycocyanins have been crystallized in the trimeric assembly form. Given that no oligomers other than monomers ($\alpha\beta$)₁, dimers ($\alpha\beta$)₂, trimers ($\alpha\beta$)₃, and hexamers ($\alpha\beta$)₆ had been described, this is not so much a “belief” as it was “these are the only known assembly forms.” The authors do not yet know the significance of the octomers ($\alpha\beta$)₈ detected, only that they can occur.
5. Line 63. Stacked, not pile. Or, stacks of...
6. General. The authors do not state the growth temperature optimum for *Thermoleptolyngbya* sp., but they are working at room temperature, which is far from optimal for a thermophile. Is the ($\alpha\beta$)₈ octomer an artifact of the lower temperature? The authors show that the ($\alpha\beta$)₆ hexamer is much more stable, and perhaps ($\alpha\beta$)₈ octomers are only stable at room temperature and not at the temperature at which this organism normally grows.
7. Line 106. Dissociated, not dissolved
8. Line 115. The authors appear to have the genome sequence for this organism, so they must know whether the genome encodes single genes for *cpcA* and *cpcB*, encoding the alpha and beta subunits for phycocyanin. This should be stated in the text. However, even if there are single genes, this does not mean that the two phycocyanins are identical, because they could be differently modified posttranslationally. Did the authors perform mass spectrometry on the two phycocyanins to demonstrate that the masses of the subunits in the two assembly forms were identical? More details about the identity of the two sets of subunits should be provided in the manuscript. Also, the authors do not compare the absorbance spectra, CD spectra, etc. for the two types of PC. These data should also be shown.
9. Line 120-122. The authors should cite Chang et al. (Cell Research, 25, 726-737 (2015)), who studied the structure of the pentacylindrical PBS of *Nostoc (Anabaena)* sp. PCC 7120.
10. Sweet et al. (J. Biol. Chem. 252, 8258-8260, 1977) showed that crystals of PC from *Anabaena variabilis* produced crystals of space group P6 with a dimer ($\alpha\beta$)₂ as the asymmetric unit. This could be relevant to the discussion here and should be cited.
11. Concerning assembly in vivo, aside from the effects of temperature, which the authors did not explore here, it is also possible that interactions with linker proteins in cells could modify the dynamics of the assembly process. Presumably, most of the PC ($\alpha\beta$)₆ hexamers would bind linkers and be inserted into PBS. Alternatively, there might be alternative linkers that could make use of the octomeric

form of PC (e.g., assemblies together with another linker such as CpcL). Does the genome encode a CpcL-type linker polypeptide?

Referee 2:

Reviewer #2 (Remarks to the Author):

This work finds and shows the unique oligomerization of TICPC. It is interesting and significant to light-harvesting of photosynthesis. The unique octomer of TICPC may be promising for bioengineering light-harvesting units and photosensory units.

This reviewer has the following concerns:

Line 127-128, How do the authors explain the discrepancy between 22.5 and 17.9 kDa for beta-TICPC? May be it important to explain the unique oligomerization of TICPC, when the discrepancy corresponds to a (post-translational) modification in apo-protein. Did the authors measure the mass of beta-TICPC using Mass Spectrometry (MS)? When not, why? When yes, what are the MS results? The mass difference between alpha and beta-TICPC in SDS-PAGE (Supplementary Fig. 1) is surprisingly great, so the MS results are key to elucidate the mass difference, which might be the molecular basis of the unexpected oligomerization of TICPC.

The authors' data show that the resolution of helices A and B, which are responsible to oligomerization, is lower (Supplementary Fig. 18). The authors should compare the sequences of the N-termini (1-30 aa) among CPCs, and compare the segments that have the higher B-factor (Supplementary Fig. 19) in alpha- and beta-TICPC with the related segments of other CPCs including PCs from cryptophytes. PCs from cryptophytes that are homologous to CPCs have also unique oligomerization. The N-termini of phycobiliproteins are responsible to oligomerization. The segments of higher B-factor might make the difference in oligomerization. In Supplementary Fig. 4, the helix B of alpha-TICPC contains E instead of K/R in other most alpha-CPCs. Does the exchange of acidic/basic amino acids make sense in oligomerization? Comprehensive analyses combining sequences with structures should shed some light on this unique oligomerization.

On description and discussion of CPC biosynthesis, the cited reference(s) are too old (e.g. 1998 in lines 92-94). According to the works of CPC biosynthesis since 2000, the CPC biosynthesis undergoes PCB-chromophorylation and methylation of subunits (the two steps are hardly involved in oligomerization), and then oligomerization. Namely the two post-translational modifications are the prerequisites for the (correct) oligomerization of CPCs.

Referee 3:

Reviewer #3 (Remarks to the Author):

This study shows structurally the apparently the same PC subunits can oligomerise into two separate forms, hexamers and octamers. This is novel as so far only hexamers have been characterised. However, in the absence of other data this observation is just a curiosity. the major question that is not addressed is does the octamer have any functional relevance. Without that this paper lacks significance. there are also a range of question that the data presented raise that need to be answered. The SDS gel of the complex is very underlapped. This means that the evidence that the sample is pure is very poor. The

authors also say that the higher molecule weight subunit runs at too high a mobility on the gel judged by its actual mass. Their observation is just left hanging. So why is this true? What proportion of the PC sample assembles into hexamers and what into octamers? If the hexamers are monomerised to they still get a mixture of oligomeric states when the sample is allowed to reoligomerise? Also vice versa with the octamers? The regions of the suture that change upon formation of the octamers rather than the hexamers needs a much fuller discussion. If they look at the sequence of their PC relative those that only form hexamers can any structural motifs be found that could suggest why this occurs? In other words when more gene sequences of PCs from new organisms are found can they use that information to predict those sequences that could support octamerisation? Also in vivo, linker polypeptides are important for formation of the phycobilisome rods. Is octamerisation affected by the presence of linkers? if so then maybe the octamers can only form in the absence of linkers?

<To Reviewer 1>

Thank you very much for your valuable comments and high evaluation of our results. We have carefully considered all your comments and revised the manuscript. We believe that the corrections and additions incorporated in the revised manuscript are appropriate. Please confirm the following responses. Thanks to your comments, we think that our manuscript has been much improved.

Comment 1

1. The authors refer to this protein as CPC, in which the first C means “cyanobacterial.” It really is unnecessary to use this letter and throughout the protein could be referred to more simply as PC, phycocyanin.

Response

As you pointed out, CPC means cyanobacterial phycocyanin as mentioned in line 78, while it is necessary to distinguish our PCs from other PCs in the manuscript because we mentioned PCs from red alga (RPC) and cryptophytes for comparison. Therefore, we utilized the term CPC to avoid misunderstanding and to emphasize that the unique octameric structure was discovered from the cyanobacterial PC. Moreover, most of scientists studying cyanobacterial PCs utilized the term CPC, so we think it is not unnatural to follow them.

Comment 2

2. Line 43. Three dimensional objects such as phycocyanin hexamers are not well-described as two-dimensional objects (e.g., a circle). Phycocyanin is shaped like a bagel; in other words, it is a toroid.

Response

As you pointed out, the 3D shapes of proteins were not well-described. Therefore, the following sentences were modified in the revised manuscript.

“...a closed toroidal hexamer...” (see page 3, line 5)

“...CPC is a closed toroidal $(\alpha\beta)_6$ hexamer...” (see page 4, line 19)

“...to give an extended closed toroidal structure...” (see page 9, line 24)

“...due to the extended toroidal structure of *TICPC-8*...” (see page 10, line 7)

Comment 3

3. Line 43-44. Delete ...although...determined.

Response

According to the comment, the description "although...determined" in line 43 was deleted.

Comment 4

4. Line 51-52. It is difficult to say if there was/is a belief that phycocyanin only assembles into hexamers. It was well-established that monomers and trimers also occur, and some phycocyanins have been crystallized in the trimeric assembly form. Given that no oligomers other than monomers $(\alpha\beta)_1$, dimers $(\alpha\beta)_2$, trimers $(\alpha\beta)_3$, and hexamers $(\alpha\beta)_6$ had been described, this is not so much a “belief” as it was “these are the only known assembly forms.” The authors do not yet know the significance of the octomers $(\alpha\beta)_8$ detected, only that they can occur.

Response

As you pointed out, it is not appropriate to use the term “belief” in this context. According to the comment, the following sentence was modified in the revised manuscript.

“These observations provide new insights into the assembly processes of CPCs and the mechanism of energy transfer in the light-harvesting complexes.” (see page 3, line 12–14)

Comment 5

5. Line 63. Stacked, not pile. Or, stacks of...

Response

Thank you for pointing out a wrong choice of word. We modified the sentence as follows.

“...whereas the rod consists of stacked phycocyanins (PCs)...” (see page 4, line 1)

Comment 6

6. General. The authors do not state the growth temperature optimum for *Thermoleptolyngbya* sp., but they are working at room temperature, which is far from optimal for a thermophile. Is the $(\alpha\beta)_8$ octomer an artifact of the lower temperature? The authors show that the $(\alpha\beta)_6$ hexamer is much more stable, and perhaps $(\alpha\beta)_8$ octomers are only stable at room temperature and not at the temperature at which this organism normally grows.

Response

The optimum temperature and growth conditions for O-77 were described in the previous report as mentioned in line 424 and references 35 and 36; these reports illustrated that O-77 could grow well in a temperature range between 25°C and 60°C with the optimal growth temperature of 55°C, and the growth temperature of O-77 in this study was at around 42°C. Slightly low growth temperature was partly due to the large-scale cultivation.

Now, we would like to emphasize that hexameric and octameric CPC structures can be observed only by X-ray crystallographic and cryo-EM analyses because CPCs are disassembled into the smaller oligomers and/or monomers in diluted solutions, which causes the difficulties in separation of hexamer and octamer together with investigating functions of novel octamers. We performed the SEC-MALS measurement of concentrated CPC solutions to reveal the solution states of hexamer and octamer, where the concentration was the same as crystallization and cryo-EM conditions. However, as the solutions were diluted during pathing through the column, we could not observe hexameric and octameric states in solution. In the same way, CPC oligomers were disassembled into the smaller oligomers and/or monomers during the isolation processes because we utilized four columns to purify CPC. Therefore, hexamers and octamers observed by X-ray crystallographic and cryo-EM analyses in this study were assembled from the smaller oligomers and/or monomers AFTER isolation. Based on these results, we think that octamers are formed even at the lower temperature (25°C) and are not artifacts of the lower temperature. In addition, we concluded the growth temperature and the assembled structure are hardly correlated with each other. Instead, we performed CD measurement of solutions prepared by dissolving crystals of hexamers or octamers to assess the thermostability of the smaller oligomers and/or monomers derived from hexamers and octamers.

We performed 1) crystallization of CPCs, 2) X-ray crystallographic analyses to confirm the reproducibility of the formation of hexameric and octameric structures, 3) picking up these two types of single crystals separately, and 4) preparing two CPC solutions. The X-ray crystallographic analyses of the single crystals showed both hexamers and octamers were crystallized from the same CPC solutions, showing that crystallization of hexamers and octamers were successfully reproduced (Figure A). The CD spectra of monomers from hexamers or octamers showed almost the same ellipticity in a range of 190–700 nm (Figure B). Thermostabilities of monomers from hexamers or octamers were investigated by measuring CD at 222 nm in a range of 20–90°C (Figure C). The thermal denaturation midpoints (T_m) were 71°C for both hexamers and octamers, showing that the physical properties of disassembled structures from hexamers and octamers were the same.

Therefore, we added the following sentences and Figure A–C as Supplementary Fig. 12 and 14 in

the revised manuscript.

“As the crystal shapes of *TICPC-6* and *TICPC-8* were different from each other (Supplementary Fig. 12), CPC solutions derived from hexameric ($\alpha\beta$)₆ and octameric ($\alpha\beta$)₈ states could be prepared by picking up crystals and dissolving them separately.” (see page 11, line 21–23)

“In addition, CD spectra of these solutions also showed almost the same ellipticity in a range of 190–250 nm (Supplementary Fig. 14a), and the thermal denaturation midpoints were 71°C for both solutions (Supplementary Fig. 14b). It should be noted that oligomeric structures were disassembled in these solutions because the concentration was low^{28–32}. These results strongly supported that the subunits composition and physical properties of *TICPC-6* and *TICPC-8* were the same and post-translational modifications did not occur except for the methylation and PCB chromophorylation (Supplementary Fig. 5 and 6). Crystals seemed to be grown one type of morphology in the same drop, and the residual crystallization solution were much clearer than the initial states (Supplementary Fig. 12), indicating the equilibrium between hexameric ($\alpha\beta$)₆ and octameric ($\alpha\beta$)₈ states during the crystallization process.” (see page 12, line 2–13)

Figure A. Photos of representative single crystals.

Figure B. CD spectra of the CPC solutions prepared by dissolving crystals of hexamer or octamer.

Figure C. Temperature dependences of CD at 222 nm for the CPC solutions prepared by dissolving crystals of hexamer or octamer.

Comment 7

7. Line 106. Dissociated, not dissolved

Response

As mentioned in the response 6, the original CPC solution contained the smaller oligomers and/or monomers whereas the concentrated CPC solution contained hexamers and octamers, thus we could not determine whether dimer was assembled from monomers or disassembled from oligomers. The analytical ultracentrifugation showed the solution mainly contained dimers, so the following sentence was modified to avoid misunderstanding.

“...analytical ultracentrifugation of the *TICPC* solution showed that the main dissolved species was an unusual dimeric ($\alpha\beta$)₂ state.” (see page 5, line 20)

Comment 8

8. Line 115. The authors appear to have the genome sequence for this organism, so they must know whether the genome encodes single genes for *cpcA* and *cpcB*, encoding the alpha and beta subunits for phycocyanin. This should be stated in the text. However, even if there are single genes, this does not mean that the two phycocyanins are identical, because they could be differently modified posttranslationally. Did the authors perform mass spectrometry on the two phycocyanins to demonstrate that the masses of the subunits in the two assembly forms were identical? More details about the identity of the two sets of subunits should be provided in the manuscript. Also, the authors do not compare the absorbance spectra, CD spectra, etc. for the two types of PC. These data should also be shown.

Response

According to the comment, firstly we revisited the genome of O-77 and confirmed that the genome encoded single *CpcA* (BAU42407) and *CpcB* (BAU42408) genes. As mentioned in the response 6, we performed the crystallization of CPC, followed by picking up them for the CD measurement. So, next, crystals of hexamers or octamers were dissolved in 10 mM phosphate buffer (pH 7.0) and the MALDI TOF mass measurement was performed by using sinapic acid as a matrix (Figure D). As a result, both solutions showed essentially the same signals: The signals at m/z 18137 and 18138 were assignable to the α subunit of *TICPC* with a PCB (calcd. 18130.29), and the signal at m/z 19260 and 19264 were assignable to the β subunit of *TICPC* with two PCBs and one methylated β Asn72 (calcd. 19249.87), strongly indicating that the subunit composition of hexamer and octamer are the same and post-translational modifications did not occur except for methylation and PCB chromophorylation. As mentioned in the response 6, the CD spectra of monomers from hexamers or octamers showed almost the same ellipticity (Figure B), Based on these results, we concluded that α and β subunits of hexamers and octamers were the same composition. Therefore, we added the following sentences and figure D as

Supplementary Fig. 13 in the revised manuscript.

“The genome encodes single genes for *CpcA* (BAU42407) and *CpcB* (BAU42408), encoding the α and β subunits for CPC.” (see page 6, line 8–9)

“Matrix-assisted laser desorption/ionization time-of-flight (MALDI-TOF) mass spectrometric analysis of these two solutions showed essentially the same two signals assignable to the α and β subunits of *TICPC* (Supplementary Fig. 13).” (see page 11, line 24–page 12, line 2)

“MALDI-TOF mass spectrometric analysis of CPC solutions were performed using a microflex LT (Bruker Daltonics, Billerica, MA, USA). The saturated sinapinic acid solution in ethanol was deposited onto a ground steel MALDI target plate to form a matrix layer. A mixture of CPC and sinapinic acid in a mixed solvent of acetonitrile and 0.1% trifluoroacetic acid (3/7, v/v) was then dropped on the matrix layer and dried at room temperature (ca. 25°C).” (see page 21, line 18–24)

Figure D. MALDI-TOF mass spectra of **a** Solution of single crystals of *TICPC*-6 and **b** solution of single crystals of *TICPC*-8. The signals at m/z 18137 and 18138 were assignable to the α subunit of *TICPC* with a PCB (calcd. 18130.29), and the signal at m/z 19260 and 19264 were assignable to the β subunit of *TICPC* with two PCBs and one methylated β Asn72 (calcd. 19249.87).

Comment 9

9. Line 120-122. The authors should cite Chang et al. (*Cell Research*, 25, 726-737 (2015)), who studied the structure of the pentacylindrical PBS of *Nostoc (Anabaena)* sp. PCC 7120.

Response

According to the comment, we replaced the reference 41 with the newer reference Chang, L. et al. Structural organization of an intact phycobilisome and its association with photosystem II. *Cell Res.* 25, 726–737 (2015).

Comment 10

10. Sweet et al. (*J. Biol. Chem.* 252, 8258-8260, 1977) showed that crystals of PC from *Anabaena variabilis* produced crystals of space group P6 with a dimer ($\alpha\beta$)₂ as the asymmetric unit. This could be relevant to the discussion here and should be cited.

Response

As you pointed out, Sweet et al. reported the hexameric CPC crystal with a dimer as the asymmetric unit. However, this crystallographic result did not mean that hexameric structures can be disassembled

into dimers in solution. In addition, we cannot compare the 3D structure of this CPC because this reference is too old and just a preliminary result. Therefore, we cannot cite this reference in our manuscript.

Comment 11

11. Concerning assembly *in vivo*, aside from the effects of temperature, which the authors did not explore here, it is also possible that interactions with linker proteins in cells could modify the dynamics of the assembly process. Presumably, most of the PC ($\alpha\beta$)₆ hexamers would bind linkers and be inserted into PBS. Alternatively, there might be alternative linkers that could make use of the octomeric form of PC (e.g., assemblies together with another linker such as CpcL). Does the genome encode a CpcL-type linker polypeptide?

Response

As you pointed out, assembly *in vivo* was not investigated in this manuscript. However, this is because hexameric and octameric CPC structures can be observed only by X-ray crystallographic and cryo-EM analyses and separation of hexamer and octamer is technically quite difficult as mentioned in the response 6. As for the linker protein, we think that the effect of linker proteins on the oligomeric states is not so important because both hexamers and octamers were crystallized from the same solution without linker proteins. Again, hexamers and octamers observed by X-ray crystallographic and cryo-EM analyses in this study were assembled from the smaller oligomers and/or monomers after isolation and purification. According to the genome, O-77 possesses 7 types of CPC linker-protein encoding genes; *CpcC1* (WP_068510405), *CpcC2* (WP_068510403), *CpcD* (WP_068510401), *CpcG4* (WP_068510398), *CpcG3* (WP_068510395), ***CpcL*** (WP_068510394), *CpcG1* (WP_068510393). Since all these linker proteins and their functions have been investigated previously (*Nature* **551**, 57–63 (2017); *Nature* **579**, 146–151 (2020); *Proc. Nat. Acad. Sci.* **111**, 2512–2517 (2014)), there are no distinctive linker proteins that is only observed in TICPC. Crystallization, structural analysis, and biochemical analysis of octameric structure with a linker protein would provide more information to investigate the functional relevance *in vivo*, but such examinations are beyond the scope of this manuscript. It should be noted that crystallization of PC with a linker protein is quite difficult because the interaction between PC and a linker protein are very weak (*Mol. Microbiol.* **82**, 698–705 (2011); *Proc. Nat. Acad. Sci.* **96**, 1363–1368 (1999)). We now try to isolate PBS from O-77 to investigate the structures of CPCs *in vivo* (Figure E). If the detail can be clarified, we will publish the results in due course.

Figure E. SDS-PAGE of PBS from O-77.

<To Reviewer 2>

Thank you very much for your valuable comments and high evaluation of our results. We have carefully considered all your comments and revised the manuscript. We believe that the corrections and additions incorporated in the revised manuscript are appropriate. Please confirm the following responses. Thanks to your comments, we think that our manuscript has been much improved.

Comment 1

Line 127-128, How do the authors explain the discrepancy between 22.5 and 17.9 kDa for beta-TICPC? May be it important to explain the unique oligomerization of TICPC, when the discrepancy corresponds to a (post-translational) modification in apo-protein. Did the authors measure the mass of beta-TICPC using Mass Spectrometry (MS)? When not, why? When yes, what are the MS results? The mass difference between alpha and beta-TICPC in SDS-PAGE (Supplementary Fig. 1) is surprisingly great, so the MS results are key to elucidate the mass difference, which might be the molecular basis of the unexpected oligomerization of TICPC.

Response

The SDS-PAGE of purified CPC from various types of organisms quite often showed bands at larger molecular weight compared with the calculated mass especially for β subunits, and thus, numerous papers showing discrepancies of bands have been reported to date. See for example; *Thermosynechococcus elongatus* (*J. Struct. Biol.* **2003**, *141*, 149.), *Phormidium rubidum* A09DM (*RSC Adv.* **2016**, *6*, 77898.), *Limnothrix* sp. NS01 (*Sci. Rep.* **2019**, *9*, 9474.), *Synechocystis aquatilis* (*J. Chromatogr. B* **2011**, *879*, 511.), *Aphanothece halophytica* (*Phytochemistry* **1991**, *30*, 3515.), *Spirulina platensis* (*Int. J. Biosci. Biochem. Bioinforma.* **2013**, *3*, 293.), *Spirulina fusiformis* (*J. Biotech.* **2003**, *102*, 55.). Therefore, this discrepancy is a normal phenomenon when estimating the mass of CPC by SDS-PAGE.

To address your comment, we performed 1) crystallization of CPCs, 2) X-ray crystallographic analyses to confirm the reproducibility of the formation of hexameric and octameric structures, 3) picking up these two types of single crystals separately, 4) preparing two CPC solutions, and 5) measuring MALDI-TOF mass spectra. It should be noted that the concentrations of solutions for hexamers (0.571 mg/mL) and octamers (0.471 mg/mL) were low enough to disassemble into the smaller oligomers and/or monomers, where the concentration of CPC solutions were determined by measuring the UV-vis spectra (Figure A). The X-ray crystallographic analyses of the single crystals showed both hexamers and octamers were crystallized from the same CPC solutions, showing that crystallization of hexamers and octamers were successfully reproduced (Figure B). The MALDI-TOF mass spectra of these solutions using sinapic acid as a matrix showed two signals assignable to calculated masses of α and β subunits of TICPC (Figure C), strongly indicating that discrepancy in SDS-PAGE was not caused by a post-translational modification of apo-protein. The discrepancy of β subunit might be explained by the higher hydrophobicity of the subunit due to two PCBs, which makes it aggregated during the SDS-PAGE and thus display higher mass in SDS-PAGE. Therefore, we added the following sentences and figure C as Supplementary Fig. 13 in the revised manuscript.

“Matrix-assisted laser desorption/ionization time-of-flight (MALDI-TOF) mass spectrometric analysis of these two solutions showed essentially the same two signals assignable to the α and β subunits of TICPC (Supplementary Fig. 13).” (see page 11, line 24–page 12, line 2)

“MALDI-TOF mass spectrometric analysis of CPC solutions were performed using a microflex LT (Bruker Daltonics, Billerica, MA, USA). The saturated sinapinic acid solution in ethanol was deposited onto a ground steel MALDI target plate to form a matrix layer. A mixture of CPC and sinapinic acid in a mixed solvent of acetonitrile and 0.1% trifluoroacetic acid (3/7, v/v) was then dropped on the matrix layer and dried at room temperature (ca. 25°C).” (see page 21, line 18–24)

Figure A. UV-vis spectra of the CPC solutions prepared by dissolving crystals of hexamer or octamer.

Figure B. Photos of representative single crystals.

Figure C. MALDI-TOF mass spectra of **a** Solution of single crystals of TICPC-6 and **b** solution of single crystals of TICPC-8. The signals at m/z 18137 and 18138 were assignable to the α subunit of TICPC with a PCB (calcd. 18130.29), and the signal at m/z 19260 and 19264 were assignable to the β subunit of TICPC with two PCBs and one methylated β Asn72 (calcd. 19249.87).

Comment 2

The authors' data show that the resolution of helices A and B, which are responsible to oligomerization, is lower (Supplementary Fig. 18). The authors should compare the sequences of the N-termini (1-30 aa) among CPCs, and compare the segments that have the higher B-factor (Supplementary Fig. 19) in alpha- and beta-TICPC with the related segments of other CPCs including PCs from cryptophytes. PCs from cryptophytes that are homologous to CPCs have also unique oligomerization. The N-termini of phycobiliproteins are responsible to oligomerization. The segments of higher B-factor might make the difference in oligomerization. In Supplementary Fig. 4, the helix B of alpha-TICPC contains E instead of K/R in other most alpha-CPCs. Does the exchange of acidic/basic amino acids make sense in oligomerization? Comprehensive analyses combining sequences with structures should shed some light on this unique oligomerization.

Response

According to the comment, we checked B-factors of reported CPC structures from various types of species. The B-factor distribution was largely dependent on the crystal packing of CPC, however, most of the CPCs showed high B-factors at the helix A_β , B_β , and G_β , which agreed well with the observation of TICPC-6 and TICPC-8 (Supplementary Fig. 22). As you pointed out, CPCs from cryptophytes are unique dimeric structures (*Proc. Nat. Acad. Sci.* **111**, E2666–E2675 (2014)), but the B-factors of these structures were largely different mainly because 1) the length of polypeptide of CPC from cyanobacteria is more than twice as long as that from cryptophytes, 2) CPCs from cryptophytes are dimers in crystal, 3) more importantly, B-factor distribution was dependent on crystal packing. Therefore, it was quite difficult to obtain meaningful information by B-factor comparison. Nevertheless, we could discuss about a possible reason to form octamer in TICPC by in-depth sequence comparison.

Firstly, all of the interaction residues at the key interfaces as shown in Figures 1g and 1h were investigated (Figure D). However, most of the residues that are associated with hydrogen bonds or salt bridges are moderately conserved compared with CPCs from other organisms (Figures E and F). Adir *et al.* reported that the hydrogen bond network between α Asp28, β Asn35, β PCB155 and α' Arg33 (α' is the α subunit of the second trimer in the hexamer) was critical to form hexameric structure in CPC from *Thermosynechococcus vulcanus* (*J. Mol. Biol.* **313**, 71–81 (2001); *Biochim. Biophys. Acta* **1827**, 311–

318 (2013)). So, next, we compared this region as shown in Figure G. The hydrogen bond network via water molecules were also observed in *TICPC*-6, and no distinctive conformational changes were observed between *TvCPC*, *TICPC*-6, and *TICPC*-8. However, *TvCPC* showed the direct hydrogen bond between β Asn35– β 155PCB, whereas *TICPC*-6 showed the indirect hydrogen bond between β Ser35– β 155PCB via a water molecule. In addition, one of the two indirect hydrogen bonds between β Asn35 and α Asp28 in *TvCPC* was not observed in *TICPC*-6. Importantly, the residue β Ser35 in *TICPC* is not well conserved among the previously reported CPC crystals. The residue α Glu32 was not involved in the CPC assembly (Figure G). In the interfaces to the α or β subunits of *TICPC*, α Glu32, α Ala72, and β Lys150 are also unique residues that are rarely observed in other reported CPC crystals (Figures E and F). The indirect hydrogen bond network via water molecules was observed between α Lys32 and β 153PCB in *TvCPC*, whereas α Glu32 was interacted with another β 153PCB in *TICPC* although the hydrogen bond was not so strong due to the multiple conformation of α Glu32 (Figure H). This observation indicated that α Lys32 in *TvCPC* is important to form ($\alpha\beta$) monomer, whereas α Glu32 in *TICPC* rather stabilizes a dimeric structure ($\alpha\beta$)₂ because α Glu32 is a key residue in this interface (Supplementary Fig. 27a). As for α Ala72, the direct hydrogen bond was observed between α Ser72 and α 84PCB in *TvCPC*, whereas no hydrogen bonds were observed between α Ala72 and α 84PCB in *TICPC* (Figure I). There were no interactions in β Pro150 of *TvCPC* and in β Lys150 of *TICPC* (Figure J). In summary, since the subunits of *TICPC* can become flexible by weakening the hydrogen bond network that is important to form hexameric structure due to β Ser35, both hexameric and octameric structures could presumably be observed. In addition, unusual α Glu32 might act as important role to stabilize dimeric ($\alpha\beta$)₂ structure.

Therefore, we added the following sentences and figures G and H as Supplementary Fig. 26.

“In addition, Adir *et al.* reported that the hydrogen bond network between α Asp28, β Asn35, and β PCB153 was critical to form hexameric structure in *TvCPC*^{16,23}, while *TICPC* possesses β Ser35 that is rarely observed in the previously reported CPC crystals: (1) the direct hydrogen bond between β Asn35– β 153PCB was observed in *TvCPC*, whereas the indirect hydrogen bond between β Ser35– β 153PCB via a water molecule was observed in *TICPC*, and (2) one of the two indirect hydrogen bonds between β Asn35– α Asp28 in *TvCPC* was not observed in *TICPC*-6, suggesting the destabilization of hexameric structures in *TICPC* (Supplementary Fig. 26a). Moreover, the indirect hydrogen bond between α Lys32– β 153PCB in *TvCPC* is important to stabilize ($\alpha\beta$) monomer, whereas α Glu32 in *TICPC* rather stabilizes a dimeric structure ($\alpha\beta$)₂ by interacting with the neighboring β subunit in the dimeric interface (Supplementary Fig. 26b and 27a). These unique residues might be responsible for unusual octamer assembly.” (see page 17, line 6–18)

Figure D. Ribbon models of key interfaces in the assembly of hexameric and octameric structures.

Figure G. Enlarged view of the hydrogen bond network.

Figure H. Enlarged view of the hydrogen bond network.

Figure I. Enlarged view of the hydrogen bond network.

Figure J. Enlarged view of βPro150 in TvCPC and βLys150 in TICPC.

Comment 3

On description and discussion of CPC biosynthesis, the cited reference(s) are too old (e.g. 1998 in lines 92-94). According to the works of CPC biosynthesis since 2000, the CPC biosynthesis undergoes PCB-chromophorylation and methylation of subunits (the two steps are hardly involved in oligomerization), and then oligomerization. Namely the two post-translational modifications are the prerequisites for the (correct) oligomerization of CPCs.

Response

As you pointed out, when describing the assembly process of CPC, references 15 and 33 are too old. Therefore, we deleted the reference 15 and replaced the reference 33 with Mimuro, M. & Kikuchi, H. Antenna Systems and Energy Transfer in Cyanophyta and Rhodophyta in *Light-Harvesting Antennas in Photosynthesis* (eds. Green, B. R. & Parson, W. W.) 281–306 (Springer, 2003) in lines 92–94. It is worth mentioning that chromophylation and methylation of subunits in TICPC were also confirmed by the X-ray crystallographic analysis (Supplementary Fig. 5 and 6) and MALDI-TOF mass spectra (Figure C).

<To Reviewer 3>

Thank you very much for your valuable comments on our manuscript. We partly agree with your comments, whereas we think that all your questions do not change our main conclusion and decrease the significance of the study. Therefore, we have carefully considered all your comments and performed several experiments to address your comments. We believe that the corrections and additions incorporated in the revised manuscript are appropriate. Please confirm the following responses. Thanks to your comments, we think that our manuscript has been much improved.

Comment 1

This study shows structurally the apparently the same PC subunits can oligomerise into two separate forms, hexamers and octamers. This is novel as so far only hexamers have been characterised. However, in the absence of other data this observation is just a curiosity. the major question that is not addressed is does the octame have any functional relevance. Without that this paper lacks significance.

Response

As you pointed out, the function of octamer has not been elucidated in this study. However, we would like to emphasize firstly that hexameric and octameric CPC structures can be observed only by X-ray crystallographic and cryo-EM analyses because CPCs are disassembled into the smaller oligomers and/or monomers in diluted solutions. In addition, there is an equilibrium between hexamers and octamers even in the concentrated solution. These features cause the difficulties in separation of hexamer and octamer, which makes it technically impossible to investigate functions of novel octamers in a solution phase. We performed the SEC-MALS measurement of concentrated CPC solutions to reveal the solution states of hexamer and octamer, where the concentration was the same as crystallization and cryo-EM conditions. However, as the solutions were diluted during pathing through the column, we could not observe hexameric and octameric states in solution. In the same way, CPC oligomers were disassembled into the smaller oligomers and/or monomers during the isolation processes because we utilized four columns to purify CPC. Based on these results, hexamers and octamers observed by X-ray crystallographic and cryo-EM analyses in this study were assembled from the smaller oligomers and/or monomers AFTER isolation. Therefore, it is technically quite difficult to prepare a solution containing only hexamer or octamer and to reveal functions of both structures. Instead, we performed CD measurement of solutions prepared by dissolving crystals of hexamers or octamers to assess the thermostability of the smaller oligomers and/or monomers derived from hexamers and octamers.

We performed 1) crystallization of CPCs, 2) X-ray crystallographic analyses to confirm the reproducibility of the formation of hexameric and octameric structures, 3) picking up these two types of single crystals separately, and 4) preparing two CPC solutions. The X-ray crystallographic analyses of the single crystals showed both hexamers and octamers were crystallized from the same CPC solutions, showing that crystallization of hexamers and octamers were successfully reproduced (Figure A). The CD spectra of monomers from hexamers or octamers showed almost the same ellipticity in a rage of 190–700 nm (Figure B). Thermostabilities of monomers from hexamers or octamers were investigated by measuring CD at 222 nm in a range of 20–90°C (Figure C). The thermal denaturation midpoints (T_m) were 71°C for both hexamers and octamers, showing that the physical properties of disassembled structures from hexamers and octamers were the same.

Importantly, we now try to isolate PBS from O-77 to investigate the structures of CPCs in vivo. We believe that it is quite significant to provide the possibility of the existence of octameric CPC structures in vivo in order to develop this research area.

Therefore, we added the following sentences and Figure A–C as Supplementary Fig. 12 and 14 in the revised manuscript.

“As the crystal shapes of *TICPC-6* and *TICPC-8* were different from each other (Supplementary Fig.

12), CPC solutions derived from hexameric ($\alpha\beta$)₆ and octameric ($\alpha\beta$)₈ states could be prepared by picking up crystals and dissolving them separately.” (see page 11, line 21–23)

“In addition, CD spectra of these solutions also showed almost the same ellipticity in a range of 190–250 nm (Supplementary Fig. 14a), and the thermal denaturation midpoints were 71°C for both solutions (Supplementary Fig. 14b). It should be noted that oligomeric structures were disassembled in these solutions because the concentration was low^{28–32}. These results strongly supported that the subunits composition and physical properties of TICPC-6 and TICPC-8 were the same and post-translational modifications did not occur except for the methylation and PCB chromophorylation (Supplementary Fig. 5 and 6). Crystals seemed to be grown one type of morphology in the same drop, and the residual crystallization solution were much clear than the initial states (Supplementary Fig. 12), indicating the equilibrium between hexameric ($\alpha\beta$)₆ and octameric ($\alpha\beta$)₈ states during the crystallization process.” (see page 12, line 2–13)

Figure A. Photos of representative single crystals.

Figure B. CD spectra of the CPC solutions prepared by dissolving crystals of hexamer or octamer.

Figure C. Temperature dependences of CD at 222 nm for the CPC solutions prepared by dissolving crystals of hexamer or octamer.

Comment 2

there are also a range of question that the data presented raise that need to be answered. The SDS gel of the complex is very underlapped. This means that the evidence that the sample is pure is very poor. The authors also say that the higher molecule weight subunit runs at too high a mobility on the gel judged by its actual mass. Their observation is just left hanging. So why is this true?

Response

The SDS-PAGE of purified CPC from various types of organisms quite often showed bands at larger molecular weight compared with the calculated mass especially for β subunits, and thus, numerous papers showing discrepancies of bands have been reported to date. See for example; *Thermosynechococcus elongatus* (*J. Struct. Biol.* **2003**, *141*, 149.), *Phormidium rubidum* A09DM (*RSC Adv.* **2016**, *6*, 77898.), *Limnothrix* sp. NS01 (*Sci. Rep.* **2019**, *9*, 9474.), *Synechocystis aquatilis* (*J. Chromatogr. B* **2011**, *879*, 511.), *Aphanothece halophytica* (*Phytochemistry* **1991**, *30*, 3515.), *Spirulina platensis* (*Int. J. Biosci. Biochem. Bioinfoma.* **2013**, *3*, 293.), *Spirulina fusiformis* (*J. Biotech.* **2003**, *102*, 55.). Therefore, this discrepancy is a normal phenomenon when estimating the mass of CPC by SDS-PAGE. In addition, we reexamined the SDS-PAGE by using more concentrated CPC solution to check the purity (Figure D), strongly indicating that there were no bands assignable to other proteins and only α and β subunits of *TICPC* were observed.

Moreover, we measured MALDI-TOF mass spectra of the CPC solutions prepared by dissolving crystals of hexamer or octamer as described in the response 1. It should be noted that the concentrations of solutions for hexamers (0.571 mg/mL) and octamers (0.471 mg/mL) were low enough to disassemble into the smaller oligomers and/or monomers, where the concentration of CPC solutions were determined by measuring the UV-vis spectra (Figure E). The MALDI-TOF mass spectra of these solutions using sinapic acid as a matrix showed two signals assignable to calculated masses of α and β subunits of *TICPC* (Figure F), strongly indicating that discrepancy in SDS-PAGE was not caused by a post-translational modification of apo-protein. The discrepancy of β subunit might be explained by the higher hydrophobicity of the subunit due to two PCBs, which makes it aggregated during the SDS-PAGE and thus display higher mass in SDS-PAGE. Therefore, we added the following sentences and figures D and F as Supplementary Fig. 1 and 13, respectively, in the revised manuscript.

“Matrix-assisted laser desorption/ionization time-of-flight (MALDI-TOF) mass spectrometric analysis of these two solutions showed essentially the same two signals assignable to the α and β subunits of *TICPC* (Supplementary Fig. 13).” (see page 11, line 24–page 12, line 2)

“MALDI-TOF mass spectrometric analysis of CPC solutions were performed using a microflex LT (Bruker Daltonics, Billerica, MA, USA). The saturated sinapic acid solution in ethanol was deposited onto a ground steel MALDI target plate to form a matrix layer. A mixture of CPC and sinapic acid in a mixed solvent of acetonitrile and 0.1% trifluoroacetic acid (3/7, v/v) was then dropped on the matrix

layer and dried at room temperature (ca. 25°C).” (see page 21, line 18–24)

Figure D. SDS-PAGE of TICPC. Lane 1, TICPC in 10 mM potassium phosphate buffer solution (pH 7.0). M, a marker.

Figure E. UV-vis spectra of the CPC solutions prepared by dissolving crystals of hexamer or octamer.

Figure F. MALDI-TOF mass spectra of **a** Solution of single crystals of TICPC-6 and **b** solution of single crystals of TICPC-8. The signals at m/z 18137 and 18138 were assignable to the α subunit of TICPC with a PCB (calcd. 18130.29), and the signal at m/z 19260 and 19264 were assignable to the β subunit of TICPC with two PCBs and one methylated β Asn72 (calcd. 19249.87).

Comment 3

What proportion of the PC sample assembles into hexamers and what into octamers? If the hexamers are monomerised to they still get a mixture of oligomeric states when the sample is allowed to reoligomerise? Also vice versa with the octamers?

Response

As mentioned in the responses 1 and 2, the composition of monomers of hexamers and octamers were the same (Figure F). In addition, the CPC solutions were diluted during the isolation process, so CPC oligomers were disassembled into the smaller oligomers and/or monomers, which means that hexamers and octamers observed by X-ray crystallographic and cryo-EM analyses in this study were assembled from the smaller oligomers and/or monomers. Figure A showed that the shapes of crystals were different; hexamers are rhombus or rod-shaped crystals, whereas octamers are cubic crystals. Figure A also showed that the shape of crystals in the same drop was one type and the residual crystallization solutions were almost clear. These results indicated that there is an equilibrium between hexamers and octamers and the equilibrium was shifted to hexamers or octamers during the crystallization process. Please note that the cryo-EM analysis revealed that this solution contained both hexamer and octamer. In summary, there are reversible disassembly/reassembly reactions between monomers, dimers, trimers, tetramers, hexamers, and octamers depending on the concentration in TICPC.

Comment 4

The regions of the suture that change upon formation of the octamers rather than the hexamers needs a much fuller discussion. If they look at the sequence of their PC relative those that only form hexamers can any structural motifs be found that could suggest why this occurs? In other words when more gene sequences of PCs from new organisms are found can they use that information to predict those sequences that could support octamerisation?

Response

According to the comment, we performed in-depth sequence comparison. Firstly, all of the interaction residues at the key interfaces as shown in Figures 1g and 1h were investigated (Figure G).

However, most of the residues that are associated with hydrogen bonds or salt bridges are moderately conserved compared with CPCs from other organisms (Figures H and I). Adir *et al.* reported that the hydrogen bond network between α Asp28, β Asn35, β PCB155 and α' Arg33 (α' is the α subunit of the second trimer in the hexamer) was critical to form hexameric structure in CPC from *Thermosynechococcus vulcanus* (*J. Mol. Biol.* **313**, 71–81 (2001); *Biochim. Biophys. Acta* **1827**, 311–318 (2013)). So, next, we compared this region as shown in Figure J. The hydrogen bond network via water molecules were also observed in *TvCPC*, and no distinctive conformational changes were observed between *TvCPC*, *TICPC*-6, and *TICPC*-8. However, *TvCPC* showed the direct hydrogen bond between β Asn35– β 155PCB, whereas *TICPC*-6 showed the indirect hydrogen bond between β Ser35– β 155PCB via a water molecule. In addition, one of the two indirect hydrogen bonds between β Asn35 and α Asp28 in *TvCPC* was not observed in *TICPC*-6. Importantly, the residue β Ser35 in *TICPC* is not well conserved among the previously reported CPC crystals. In the interfaces to the α or β subunits of *TICPC*, α Glu32, α Ala72, and β Lys150 are also unique residues that are rarely observed in other reported CPC crystals (Figures H and I). The indirect hydrogen bond network via water molecules was observed between α Lys32 and β 153PCB in *TvCPC*, whereas α Glu32 was interacted with another β 153PCB in *TICPC* although the hydrogen bond was not so strong due to the multiple conformation of α Glu32 (Figure K). This observation indicated that α Lys32 in *TvCPC* is important to form ($\alpha\beta$) monomer, whereas α Glu32 in *TICPC* rather stabilizes a dimeric structure ($\alpha\beta$)₂ because α Glu32 is a key residue in this interface (Supplementary Fig. 27a). As for α Ala72, the direct hydrogen bond was observed between α Ser72 and α 84PCB in *TvCPC*, whereas no hydrogen bonds were observed between α Ala72 and α 84PCB in *TICPC* (Figure L). There were no interactions in β Pro150 of *TvCPC* and in β Lys150 of *TICPC* (Figure M). In summary, since the subunits of *TICPC* can become flexible by weakening the hydrogen bond network that is important to form hexameric structure due to β Ser35, both hexameric and octameric structures could presumably be observed. In addition, unusual α Glu32 might act as important role to stabilize dimeric ($\alpha\beta$)₂ structure.

Therefore, we added the following sentences and figures J and K as Supplementary Fig. 26.

“In addition, Adir *et al.* reported that the hydrogen bond network between α Asp28, β Asn35, and β PCB153 was critical to form hexameric structure in *TvCPC*^{16,23}, while *TICPC* possesses β Ser35 that is rarely observed in the previously reported CPC crystals: (1) the direct hydrogen bond between β Asn35– β 153PCB was observed in *TvCPC*, whereas the indirect hydrogen bond between β Ser35– β 153PCB via a water molecule was observed in *TICPC*, and (2) one of the two indirect hydrogen bonds between β Asn35– α Asp28 in *TvCPC* was not observed in *TICPC*-6, suggesting the destabilization of hexameric structures in *TICPC* (Supplementary Fig. 26a). Moreover, the indirect hydrogen bond between α Lys32– β 153PCB in *TvCPC* is important to stabilize ($\alpha\beta$) monomer, whereas α Glu32 in *TICPC* rather stabilizes a dimeric structure ($\alpha\beta$)₂ by interacting with the neighboring β subunit in the dimeric interface (Supplementary Fig. 26b and 27a). These unique residues might be responsible for unusual octamer assembly.” (see page 17, line 6–18)

Figure G. Ribbon models of key interfaces in the assembly of hexameric and octameric structures.

Figure J. Enlarged view of the hydrogen bond network.

Figure K. Enlarged view of the hydrogen bond network.

Figure L. Enlarged view of the hydrogen bond network.

Figure M. Enlarged view of βPro150 in TvCPC and βLys150 in TICPC.

Comment 5

Also *in vivo*, linker polypeptides are important for formation of the phycobilisome rods. Is octamerisation affected by the presence of linkers? if so then maybe the octamers can only form in the absence of linkers?

Response

As you pointed out, assembly *in vivo* was not investigated in this manuscript. However, this is because hexameric and octameric CPC structures can be observed only by X-ray crystallographic and cryo-EM analyses and separation of hexamer and octamer is technically quite difficult as mentioned in the response 1. As for the linker protein, we think that the effect of linker proteins on the oligomeric states is not so important because both hexamers and octamers were crystallized from the same solution without linker proteins. Again, hexamers and octamers observed by X-ray crystallographic and cryo-EM analyses in this study were assembled from the smaller oligomers and/or monomers. According to the genome, O-77 possesses 7 types of CPC linker-protein encoding genes; *CpcC1* (WP_068510405), *CpcC2* (WP_068510403), *CpcD* (WP_068510401), *CpcG4* (WP_068510398), *CpcG3* (WP_068510395), *CpcL* (WP_068510394), *CpcG1* (WP_068510393). Since all these linker proteins and their functions have been investigated previously (*Nature* **551**, 57–63 (2017); *Nature* **579**, 146–151 (2020); *Proc. Nat. Acad. Sci.* **111**, 2512–2517 (2014)), there are no distinctive linker proteins that is only observed in TICPC. Crystallization, structural analysis, and biochemical analysis of octameric structure with a linker protein would provide more information to investigate the functional relevance *in vivo*, but such examinations are beyond the scope of this manuscript. It should be noted that crystallization of PC with a linker protein is quite difficult because the interaction between PC and a linker protein are very weak (*Mol. Microbiol.* **82**, 698–705 (2011); *Proc. Nat. Acad. Sci.* **96**, 1363–1368 (1999)). As mentioned in the response 1, we now try to isolate PBS from O-77 to investigate the structures of CPCs *in vivo* (Figure N). If the detail can be clarified, we will publish the results in due course.

Figure N. SDS-PAGE of PBS from O-77.

REVIEWERS' COMMENTS:

Reviewer #1 (Remarks to the Author):

These authors have stumbled onto something unusual, and they are correct that it is highly unusual for a protein to assemble into oligomers with 3- and 4-fold symmetry. That makes this protein relatively unique and interesting to study. They presently have no data to suggest that this happens in vivo, but this is the first step toward determining if that might be the case. I think they have done a good job of answering concerns on the whole. I now recommend publication.

While I will not make a major issue of these things, I do want to respond to the authors with a couple of specific comments. Firstly, it is not likely that the distribution of oligomeric states is the same at 55°C as it is at 25°C. Temperature could certainly play a major role in determining this distribution. Secondly, while it may be the case that both forms are observed in vitro under the same conditions, linker proteins could easily eliminate one of these forms in vivo by locking the protein selectively into one state. For example, a linker could stabilize the hexameric form which is further stabilized by inclusion in peripheral rods of the phycobilisome. This would change the equilibrium among the oligomeric states and could the distribution in the direction of hexamers while minimizing octamers in vivo. Similarly, a different linker protein could selectively stabilize the octameric state for some particular purpose. The authors do not describe how many genes are present for potential phycocyanin linker proteins, but presumably there are more than one if this organism is like other cyanobacteria that have phycobilisomes with pentacylindrical cores.

Finally: line 40. "Giant" should be deleted, as this either implies that there is something special about these phycobilisomes or conjures up images of Godzilla.

Line 429. ...firmly confirmed... is awkward and should be modified. Just delete firmly or change.

Reviewer #2 (Remarks to the Author):

The article has been substantially improved.

By the way, the structural differences between TICPC-6 and TICPC-8 may be able to be detected using FT-IR that is highly sensitive to structural changes and need only a little bit samples. As FT-IR cares only for wavenumbers instead of absorbance/transmission, so one can detect the differences in wavenumbers of the samples at different concentrations (e.g. the various concentration of protein samples can be smeared on BaF₂ crystal for FT-IR, and then difference FT-IR, corresponding to changes in hexamer/octomer of TICPC). If possible, the authors can try.

Reviewer #3 (Remarks to the Author):

The M/S is much improved. The authors have answered most of my questions and clarified their text accordingly. The only remaining issue is whether the octameric form has any functional significance. That should be clearly addressed.

Referees' comments (COMMSBIO-21-1096A)

Referee 1:

Reviewer #1 (Remarks to the Author):

These authors have stumbled onto something unusual, and they are correct that it is highly unusual for a protein to assemble into oligomers with 3- and 4-fold symmetry. That makes this protein relatively unique and interesting to study. They presently have no data to suggest that this happens in vivo, but this is the first step toward determining if that might be the case. I think they have done a good job of answering concerns on the whole. I now recommend publication.

While I will not make a major issue of these things, I do want to respond to the authors with a couple of specific comments. Firstly, it is not likely that the distribution of oligomeric states is the same at 55°C as it is at 25°C. Temperature could certainly play a major role in determining this distribution. Secondly, while it may be the case that both forms are observed in vitro under the same conditions, linker proteins could easily eliminate one of these forms in vivo by locking the protein selectively into one state. For example, a linker could stabilize the hexameric form which is further stabilized by inclusion in peripheral rods of the phycobilisome. This would change the equilibrium among the oligomeric states and could the distribution in the direction of hexamers while minimizing octamers in vivo. Similarly, a different linker protein could selectively stabilize the octameric state for some particular purpose. The authors do not describe how many genes are present for potential phycocyanin linker proteins, but presumably there are more than one if this organism is like other cyanobacteria that have phycobilisomes with pentacylindrical cores.

Finally: line 40. "Giant" should be deleted, as this either implies that there is something special about these phycobilisomes or conjures up images of Godzilla.

Line 429. ...firmly confirmed... is awkward and should be modified. Just delete firmly or change.

Referee 2:

Reviewer #2 (Remarks to the Author):

The article has been substantially improved.

By the way, the structural differences between TICPC-6 and TICPC-8 may be able to be detected using FT-IR that is highly sensitive to structural changes and need only a little bit samples. As FT-IR cares only for wavenumbers instead of absorbance/transmission, so one can detect the differences in wavenumbers of the samples at different concentrations (e.g. the various concentration of protein samples can be smeared on BaF2 crystal for FT-IR, and then difference FT-IR, corresponding to changes in hexamer/octomer of TICPC). If possible, the authors can try.

Referee 3:

Reviewer #3 (Remarks to the Author):

The M/S is much improved. The authors have answered most of my questions and clarified their text accordingly. The only remaining issue is whether the octameric form has any functional significance. That should be clearly addressed.

<To Reviewer 1>

Thank you very much for your valuable comments and high evaluation of our revised manuscript. We have carefully considered all your comments and revised the manuscript. We believe that the corrections and additions incorporated in the revised manuscript are appropriate. Please confirm the following responses. Thanks to your comments, we think that our manuscript has been much improved.

Comment 1

Firstly, it is not likely that the distribution of oligomeric states is the same at 55°C as it is at 25°C. Temperature could certainly play a major role in determining this distribution.

Response

According to the comment, we performed crystallization of CPC at 42°C and 55°C. As a result, cubic crystals assignable to octamers were obtained at 42°C and 55°C (Figure A). Since both hexamers and octamers were also observed at high temperature, we think that oligomeric states cannot be controlled only by changing ambient temperature. In addition, the octameric structure is stable at the optimal growth temperature of 55°C, indicating that the existence of octameric structure in vivo cannot be denied. Based on the results, the following sentence was added to the revised manuscript.

“The cubic crystals assignable to octamers were also obtained at 55°C which is the optimal growth temperature of O-77.” (page 23, line 14–15)

Figure A. Photos of crystals assignable to octamers at the high temperature.

Comment 2

Secondly, while it may be the case that both forms are observed in vitro under the same conditions, linker proteins could easily eliminate one of these forms in vivo by locking the protein selectively into one state. For example, a linker could stabilize the hexameric form which is further stabilized by inclusion in peripheral rods of the phycobilisome. This would change the equilibrium among the oligomeric states and could the distribution in the direction of hexamers while minimizing octamers in vivo. Similarly, a different linker protein could selectively stabilize the octameric state for some particular purpose. The authors do not describe how many genes are present for potential phycocyanin linker proteins, but presumably there are more than one if this organism is like other cyanobacteria that have phycobilisomes with pentacylindrical cores.

Response

As you pointed out, linker proteins play an important role to stabilize hexameric structures in vivo. We stated in the response 11 of the previous point-by-point response that O-77 possesses 7 types of

CPC linker-protein encoding genes; *CpcC1* (WP_068510405), *CpcC2* (WP_068510403), *CpcD* (WP_068510401), *CpcG4* (WP_068510398), *CpcG3* (WP_068510395), *CpcL* (WP_068510394), *CpcG1* (WP_068510393). Since all these linker proteins and their functions have been investigated previously (*Nature* **551**, 57–63 (2017); *Nature* **579**, 146–151 (2020); *Proc. Nat. Acad. Sci.* **111**, 2512–2517 (2014)), there are no distinctive linker proteins that is only observed in *TICPC*. However, it is still unclear that these linker proteins can associate with octamers, so we will focus on the biological significance of the octameric structure by studying PBS of O-77. According to the comment, the following sentence was added to the revised manuscript.

“Although the biological significance of the octameric structure remains to be clarified, we will focus on it in future studies.” (page 19, line 21–22)

Comment 3

Finally: line 40. "Giant" should be deleted, as this either implies that there is something special about these phycobilisomes or conjures up images of Godzilla.

Response

According to the comment, the term "giant" in line 40 was deleted.

Comment 4

Line 429. ...firmly confirmed... is awkward and should be modified. Just delete firmly or change.

Response

According to the comment, the term "firmly" in line 429 was deleted.

<To Reviewer 2>

Thank you very much for your valuable comment and high evaluation of our revised manuscript. We have carefully considered your comment. Please confirm the following response.

Comment 1

By the way, the structural differences between TICPC-6 and TICPC-8 may be able to be detected using FT-IR that is highly sensitive to structural changes and need only a little bit samples. As FT-IR cares only for wavenumbers instead of absorbance/transmission, so one can detect the differences in wavenumbers of the samples at different concentrations (e.g. the various concentration of protein samples can be smeared on BaF2 crystal for FT-IR, and then difference FT-IR, corresponding to changes in hexamer/octomer of TICPC). If possible, the authors can try.

Response

According to the comment, we tried to isolate crystals of hexamers and octamers as solids to measure FT-IR. However, we could not measure IR spectra because 1) crystals were very fragile to collect and 2) the amount of crystals collected were not enough to obtain meaningful spectra.

<To Reviewer 3>

Thank you very much for your high evaluation of our revised manuscript. We have carefully considered your comment and revised the manuscript. We believe that the addition incorporated in the revised manuscript is appropriate. Please confirm the following responses. Thanks to your comments, we think that our manuscript has been much improved.

Comment 1

The only remaining issue is whether the octameric form has any functional significance. That should be clearly addressed.

Response

As you pointed out, the function of octamer has not been elucidated in this study. However, this is because isolation of octamer in vitro and observation of octamer in vivo are quite difficult. One might suggest that octamers can play roles to form unique PBS or to form as intermediate structure of hexamers under specific conditions because O-77 lives in the hot spring with stream under strong ambient light. However, elucidating the function of octamers is quite difficult and are beyond the scope of this manuscript. We now try to isolate PBS from O-77 to investigate the structures of CPCs in vivo, and we believe that it is quite significant to provide the possibility of the existence of octameric CPC structures in vivo in order to develop this research area. According to the comment, the following sentence was added to the revised manuscript.

“Although the biological significance of the octameric structure remains to be clarified, we will focus on it in future studies.” (page 19, line 21–22)